



# Driving mechanisms of the dissolved oxygen budget in the Levantine Sea: a coupled physical-biogeochemical modelling approach

Joelle Habib[1,2,3], Caroline Ulses[1], Claude Estournel[1], Milad Fakhri[3], Patrick Marsaleix[1], Thierry Moutin[4], Dominique Lefevre[4], Mireille Pujo-Pay[5], Marine Fourrier[2], Laurent Coppola[2,6], Cathy Wimart-Rousseau[7] and Pascal Conan[5,6†]

[1]Laboratoire d'Etudes en Géophysique et Océanographie Spatiales (LEGOS), Université de Toulouse, CNES/CNRS/IRD/UT3, 14 avenue Edouard Belin, 31400 Toulouse, France

[2]Sorbonne Université, CNRS, Laboratoire d'Océanographie de Villefranche, LOV, 06230 Villefranche-sur-Mer, France

[3]National Center for Marine Sciences, National Council for Scientific Research (CNRS-L), Jounieh, Lebanon

[4] Aix Marseille Univ, Université de Toulon, CNRS, IRD, MIO, Marseille, France

[5]Laboratoire d'Océanographie Microbienne (LOMIC), CNRS, UMR 7621, Sorbonne Université, 1 Avenue Pierre Fabre, 66651 Banyuls-sur-mer, France

[6]Sorbonne Université, CNRS OSU STAMAR – UAR2017, 4 Place Jussieu, 75252 Paris, France

[7]National Oceanography Centre Southampton, European Way, Southampton, SO14 3ZH, UK

[†] Deceased

*Correspondence to*: Joelle Habib (joellehabib22@hotmail.com)

**Abstract.**

The Levantine Basin is an ultra-oligotrophic region and the formation site of the Levantine Intermediate Waters. For the first time, a high-resolution 3D coupled hydrodynamic-biogeochemical model, SYMPHONIE-Eco3MS, was used to investigate the seasonal and interannual variability of dissolved oxygen ($O_2$) in the Levantine Basin and estimate its basin-wide budget for the period 2013–2020. Our results show that the simulated $O_2$ concentrations align well with *in situ* data from research cruises and Argo floats. During winter, the surface layer is undersaturated in oxygen by up to 2% across the entire basin, leading to atmospheric oxygen absorption. The model shows that on an annual scale, the basin acts as a net sink for atmospheric oxygen, with the Rhodes Gyre exhibiting uptake rates twice as high as the rest of the Levantine Basin. The surface layer also serves as a source of dissolved oxygen for intermediate depths, with $4.2 \pm 1.1$ mol m$^{-2}$ year$^{-1}$ of dissolved oxygen vertically transported. Oxygen is transported laterally into the basin from the Ionian Sea and exported towards the Aegean Sea, with winter heat loss intensity enhancing this lateral export at both surface and intermediate layers. The Levantine Basin alternates between autotrophic and heterotrophic states, depending on the intensity of winter surface heat loss. Spatially, the Rhodes Gyre emerges as a significant oxygen pump, contributing 41% of the total oxygen production in the surface layer in the Levantine basin. This study highlights the need for further modeling studies on pluri-annual and multi-decadal scales to explore the interannual



variability and evolution of the annual oxygen budget across the entire Eastern Basin, particularly in the context of
climate change.

## 1 Introduction

Dissolved oxygen ($O_2$) is essential for marine life, supporting respiration of living organisms and the oxidation of
organic matter, and therefore influencing the biogeochemical cycles of important elements in the ocean. The ocean's
oxygen inventory is primarily controlled by photosynthesis, respiration of organic matter, remineralization,
temperature and salinity-dependent oxygen solubility, air-sea exchange and the mixing and advective fluxes
influencing the ventilation of water masses. Since 1960, the total oxygen inventory has decreased by 2% in the Global
Ocean (Schmidtko et al., 2017). This decrease in oxygen inventory referred to as deoxygenation has been attributed
to the global warming which leads to the reduction of oxygen solubility, explaining ~15% of the current total global
oxygen loss (Schmidtko et al., 2017), and the increase of upper ocean stratification generating a reduction of
ventilation and circulation of deep ocean layers (Helm et al., 2011; Schmidtko et al., 2017). However, oxygen changes
present large regional and temporal variability (Schmidtko et al., 2017; Stramma and Schmidtko, 2021). The decline
in oxygen inventory induced by global warming, combined with seasonal, interannual, and decadal variations, is barely
detectable in the upper layer of the water column. The description and assessment of the various drivers and processes
influencing oxygen changes may provide valuable insight into the complex regional variations in oxygen dynamics.
The objective of the present work is to assess the main mechanisms controlling the oxygen changes and budget in the
ultra-oligotrophic south-eastern Mediterranean Sea at seasonal and interannual time scales over 7 years. The
Mediterranean Sea is a well-ventilated basin compared to the World Ocean (Schneider et al., 2014; Tanhua et al.,
2013). The low salinity Atlantic Water (AW) enters the surface layer through the Gibraltar Strait and travels towards
the eastern sub-basin. It gradually transforms into a more saline Mediterranean Water due to the air-sea heat and
moisture fluxes (Malanotte-Rizzoli et al., 2014). Reaching the Levantine Basin (Fig. 1), the Modified AW (MAW)
subducts under the warm and saline Levantine Surface Water (LSW) in summer. During winter, surface heat loss
increases water density, triggering vertical mixing that reaches intermediate depths and leads to the formation of the
Levantine Intermediate Water (LIW). The Rhodes Gyre, a permanent cyclone in the northwest of the Levantine Basin
(Lascaratos et al., 1999; Lascaratos and Nittis, 1998; Sur et al., 1993), has been identified as the main area of LIW
formation. Then, LIW propagates between 200-600 m in the whole Mediterranean (Brasseur et al., 1996). This water
impacts the deep water formation in both the Adriatic Sea (Gačić et al., 2010; Lascaratos et al., 1999) and the Gulf of
Lion (Schneider et al., 2014), acting as a preconditioning factor for these water formations with its high salinity. Part
of LIW also exits the Mediterranean Sea, flowing toward the Atlantic.
The Levantine Basin is considered an ultra-oligotrophic region, except the Rhodes Gyre, where winter vertical
convection enriches the surface layer with nutrients, stimulating organic carbon production (D'Ortenzio et al., 2021;
Lavigne et al., 2013) and its export to the intermediate and surrounding areas (Habib et al., 2023). Previous studies
have investigated the oxygen distribution and dynamics in this region (Di Biagio et al., 2022; Klein et al., 2003; Manca
et al., 2004; Mavropoulou et al., 2020; Schlitzer et al., 1991; Tanhua et al., 2013). The vertical distribution in the basin
is characterized by a surface layer exhibiting seasonal variability. During the stratified period, the upper waters present
an oversaturated surface water and a sub-surface oxygen maximum (235 $\mu$mol kg$^{-1}$) between 50-80 m (Di Biagio et
al., 2022; Kress et al., 2003; Mavropoulou et al., 2020), attributed to various processes such as primary production



within that layer, or alternatively, downward transport mechanisms like subduction (Di Biagio et al., 2022). In winter,
the oxygen vertical profile shows an upper mixed layer with maximum dissolved concentrations of 240 µmol kg$^{-1}$,
characterised by an undersaturation in oxygen at the surface related to the atmosphere. At intermediate depths, the
LIW oxygen has values ranging from 197 to 210 µmol kg$^{-1}$ (Mavropoulou et al., 2020). An Oxygen Minimum Layer
(OML) is located between 600-1200 m with a minimum concentration of 170/180 µmol kg$^{-1}$ (Tanhua et al., 2013).
Below this layer, deep waters originating from the Adriatic and Aegean seas present values above 185 µmol kg$^{-1}$,
slightly higher than the ones recorded for the OML (Mavropoulou et al., 2020).
The Levantine Basin shows spatial changes in oxygen content occurring at short, annual, and decadal time scales
(Kress et al., 2014; Sisma-Ventura et al., 2016). The modeling study by Cossarini et al. (2021) showed a negative
trend in oxygen concentration at the surface of the Mediterranean Sea due to the surface temperature increase over the
past two decades (Escudier et al., 2021; Ozer et al., 2016; 2022). Mavropoulou et al. (2020) highlighted a variability
in deep and intermediate layers' oxygen concentration linked to shifts in the formation location of water masses
between the Adriatic and the Aegean seas. In particular, in the 1990s, during the Eastern Mediterranean Transient
(EMT), warmer, saltier, and more oxygenated waters originating from the Aegean Sea flowed into the deep layers of
the Levantine Basin (Lascaratos et al., 1999). A net decrease in the oxygen inventory in deeper layers (1200-2000 m)
of the southeastern Levantine Basin over the 20 years from 2002 to 2020 has also been pointed out by Sisma-Ventura
et al. (2021), reflecting a return to pre-EMT state due to mixing between Aegean and Adriatic waters. To date, the
oxygen inventory in the Eastern Mediterranean Basin remains poorly understood, with limited spatial and temporal
observations in the Levantine Basin, and there is no proposed comprehensive budget quantification for the entire
region. The PERLE (Pelagic Ecosystem Response to Deep Water Formation in the Levant Experiment) project aimed
to gain insights into the biogeochemical cycles in this region through multi-platform observations and modelling. In
this study, we quantify the seasonal and interannual variations in the oxygen inventory of the Levantine surface and
intermediate water masses, detailing the contribution of air-sea oxygen fluxes, biological and physical processes. This
analysis is based on 3D coupled hydrodynamic-biogeochemical model outputs covering a period of 7 years, from 2013
to 2020.
After the introduction (Sect. 1), this paper is organized as follows. Sect. 2 describes the coupled hydrodynamic-
biogeochemical model implemented in the Levantine Basin and observations used for the model assessment. Sect. 3
first presents an assessment of the model results using *in situ* observations, then investigates the seasonal and
interannual dynamics of oxygen in the surface and intermediate layers for the Levantine Basin, describes its spatial
variability, and finally estimates an annual budget of oxygen. This section is followed by a discussion of the results
and a conclusion in Sect. 4 and 5, respectively.
**2. Material and Method**
**2.1 Modeling**
**2.1.1 The coupled hydrodynamic-biogeochemical model**
The modeling presented in this study is based on the biogeochemical model Eco3M-S forced offline by the ocean
circulation model SYMPHONIE, described in detail in Marsaleix et al. (2006, 2008), Estournel et al. (2016), and
Damien et al. (2017). This latter is a 3D primitive equation model with a free surface and generalized sigma vertical





coordinate previously used to simulate the hydrodynamic conditions of the Mediterranean Sea: in river plumes
(Estournel et al., 1997, 2001; Marsaleix et al., 1998), for dense water formation (Estournel et al., 2005, 2016;
Herrmann et al., 2008; Ulses et al., 2008) and shelf-slope exchanges (Mikolajczak et al., 2020).
We used the biogeochemical model Eco3M-S, a multi-nutrient and multi-plankton functional type model, representing
the dynamics of the pelagic plankton ecosystem and the cycles of carbon, nitrogen, phosphorus, silicon, and oxygen
(Auger et al., 2011; Ulses et al., 2023). The phytoplankton is represented by three size classes: pico-, nano-, and micro-
phytoplankton (named class 1, 2, and 3, respectively), with variable internal ratios. The zooplankton is also represented
by three size classes: nano-, micro-, and mesozooplankton (named class 1, 2, and 3, respectively). A compartment of
bacteria has been explicitly taken into account. The internal composition varies for phytoplankton and remains
constant for heterotrophic organisms. Four dissolved inorganic nutrients have been considered: nitrate, ammonium,
phosphate, and silicate. In addition to dissolved organic matter (DOM), particulate organic matter (POM) has been
divided into two weight classes, namely light and heavy. The biogeochemical model was previously used to study the
dynamics of the planktonic ecosystems and organic carbon (Auger et al., 2014; Herrmann et al., 2013; Kessouri et al.,
2018; Many et al., 2021; Ulses et al., 2016), as well as the nitrogen and phosphorus cycles (Kessouri et al., 2017), and
the oxygen dynamics (Ulses et al., 2021) in the northwestern Mediterranean Sea. The rate of change of dissolved
oxygen concentration due to biogeochemistry in the water column is calculated based on the following equation:
$$\frac{dDOx}{dt} = \sum_{i=1}^{3}(GPP_i - RespPhy_i)\gamma_{C/DOx} - \sum_{i=1}^{3}(RespZoo_i)\gamma_{C/DOx} - RespBac\,\gamma_{C/DOx}$$

$$+ (UptPhy_{\,i,NO_3} - Nitrif)\gamma_{NH_4/DOx} \qquad \text{(Eq. 1)}$$


The dissolved oxygen concentration is represented by the term $DOx$. $GPP_i$ and $RespPhy_i$ are gross primary production
and respiration, respectively, for phytoplankton group i. $RespZoo_i$ and $RespBac$ are respiration of zooplankton group
i and of bacteria, $UptPhy_{i,NO3}$, and Nitrif uptake of nitrate by phytoplankton class i, and nitrification, respectively.
$\gamma_{C/DOx}$ and $\gamma_{NH_4/DOx}$ , equal to 1 and 2, respectively, are the moles of DOx used per mole of C in respiration and
needed to oxidize one mole of ammonium in nitrification as described in Grégoire et al. (2008).
The flux of dissolved oxygen at the air-sea interface is governed by the following equation:
$DOxFlux = Kw(DOxSat - DOxSurf)$   (Eq. 2)
$DOxSat$ represents the oxygen saturation also known as solubility and $DOxSurf$ the concentration of dissolved
oxygen at the surface. The dissolved oxygen at solubility level is determined using the Garcia and Gordon (1992)
equation. The oxygen saturation anomaly (OSA, expressed in percentage) is defined as the difference between the
dissolved oxygen concentration and the solubility: $(DOx - DOxSat)/DOxSat$ x100 %. $Kw$ represents the $O_2$ transfer
velocity in m s$^{-1}$. We used the parametrization of Wanninkhof and McGillis (1999) with a cubic dependency to the
10-m wind speed following the study of Ulses et al. (2021) in the northwestern Mediterranean deep convection area,
which obtained the best fits with in situ observations of oxygen concentration using this parametrization.




### 2.1.2 Implementation

The implementation of the coupled physical-biogeochemical model was described in detail in Habib et al. (2023). The hydrodynamic model covers the Mediterranean Sea and the Marmara Sea and it extends to 8° west in the Gulf of Cadiz.The horizontal resolution varies between 2.3 and 4.5 km, in general. A narrowing was conducted in the Gibraltar Strait with a 1.3 km grid for a better representation of the exchange area between the Mediterranean Sea and the Atlantic Ocean. The model has 60 vertical vanishing quasi sigma levels (Estournel et al., 2021) with closer levels near the surface. The period simulated by the hydrodynamic model runs from May 2011 to May 2021. This model configuration was used to describe the surface and intermediate water circulations in the eastern Mediterranean Sea (Estournel et al., 2021). Atmospheric forcings were provided by the ECMWF model with a horizontal resolution of 1/8° using hourly fields (wind, air temperature and humidity, pressure, solar and downward longwave radiation, and precipitation). The model accounts for a total of 142 rivers (Fig. 1).

The biogeochemical model Eco3M-S was forced by daily fields of temperature, salinity, current, and vertical diffusivity from the SYMPHONIE model. It covers the period between August 2011 till March 2021. The two first years for the biogeochemical model were considered as a spin-up and were not considered in the analysis. We used wind speed and solar radiation atmospheric forcings provided by the ECMWF model as for the hydrodynamic simulation. The biogeochemical model was initialized using climatological fields of *in situ* nutrient and dissolved oxygen concentrations from the CARIMED (CARbon in the MEDiterranean Sea, Alvarez et al., 2019) database and Biogeochemical-Argo (BGC-Argo) float data over the 2011-2012 summer periods when data were available, in 10 sub-regions. At the river mouths, concentrations of nutrients were imposed by sub-basin using the dataset of Ludwig et al. (2010). Dissolved oxygen at river mouths was set at saturation values. In the Atlantic Ocean, nutrients were prescribed using monthly profiles from the World Ocean Atlas 2009 climatology at 5.5 °W. In the Marmara Sea, to represent a two-layer flow regime, we imposed a daily relaxation towards a nutrient concentration of 0.24 and 1.03 mmol N $m^{-3}$ and a phosphate concentration of 0.06 and 0.05 mmol P $m^{-3}$ for depths above and below 15 m, respectively, based on the observations near the Dardanelles Strait from (Tugrul et al., 2002).

### 2.1.3 Study area and budget calculation

The study area (delimited by the black lines, Fig. 1) covers 540 000 $m^2$. For spatial mean and budget calculation, the water column was divided into three layers based on the biogeochemical processes as well the depth of the LIW: the surface layer defined as the photic layer covering the surface to 150 m depth where photosynthesis takes place, the underlying intermediate layer from 150 to 400 m where LIW flows, and the deep layer below 400 m (Estournel et al., 2021). In this study, we will be focusing on the first two layers, as changes at greater depths are very slow over the 8-year period and barely detectable. The biogeochemical term of the oxygen budget is the sum of oxygen production due to gross primary production and nitrate uptake by phytoplankton, and of oxygen consumption through nitrification and community respiration. The physical term is divided into two components: the lateral and the vertical transports, which are both due to advection and mixing processes. The lateral transport represents the exchanges with the Ionian and Aegean seas. A negative lateral transport indicates a net export of oxygen from the considered layer of the Levantine Basin. The oxygen inventory, air-sea fluxes, biogeochemical fluxes, and lateral fluxes were calculated online while the vertical transport, defined as a net flux at the layer interface, was deduced from the other terms of the budget. The budget calculation is detailed in Text S1 in Supplement Material.



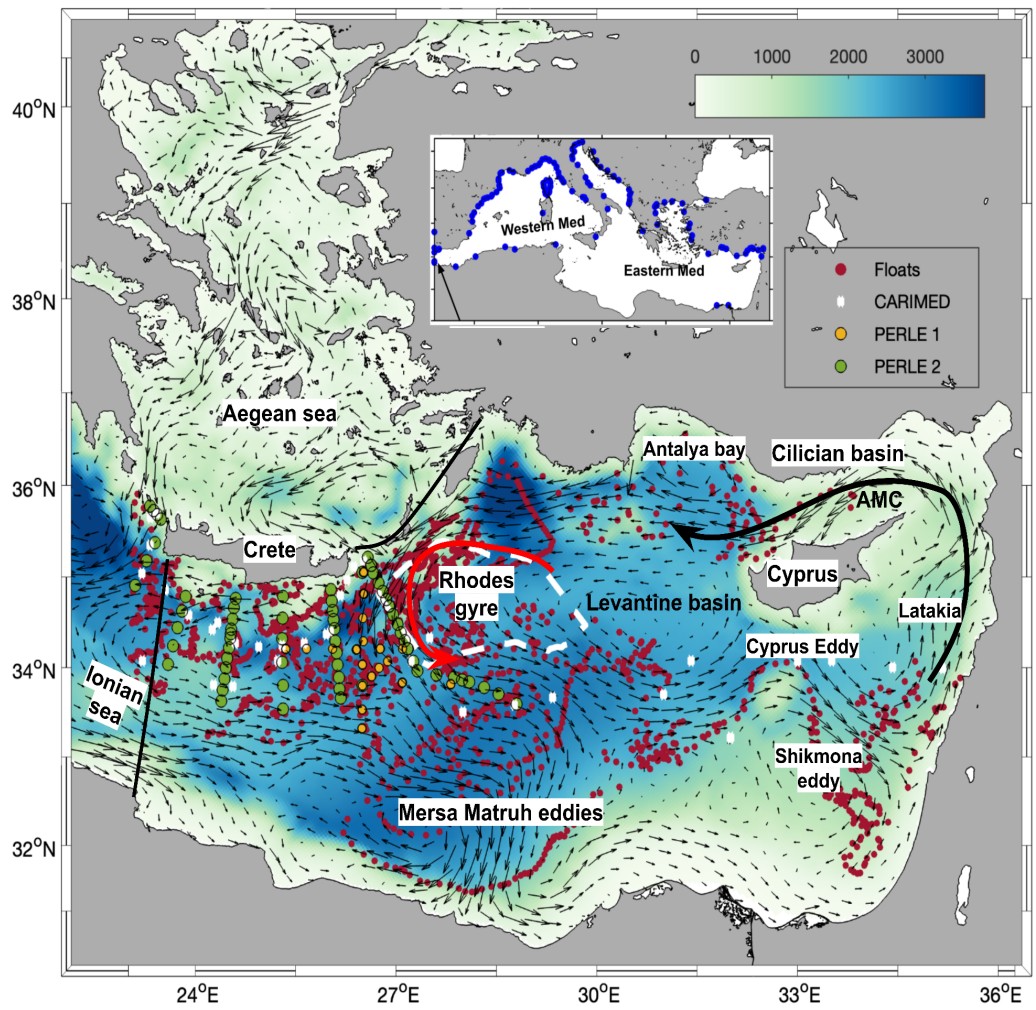

**Figure 1: Model domain and bathymetry (m, background) in the Eastern Mediterranean. The arrows represent the simulated surface currents averaged over the study 7-year period (2013-2020), black thick lines delimit the basin for the budget calculation. Red, yellow, and green dots indicate BGC-Argo floats trajectories, PERLE-1, and PERLE-2 cruise stations, respectively, and white crosses CARIMED cruise stations, over the period from 2013 to 2021. Blue dots in the insert represent the river mouths.**

## 2.2 Observations for model assessment

### 2.2.1. Argo float data

In order to assess the modeled temporal and spatial evolutions of the oxygen concentration, we use observations from Argo floats that were deployed in the Levantine Basin during the periods of 2013-2015 and 2015-2018. In particular,



we present comparisons with data provided by the two BGC-Argo floats 6901528 and 6901764 (151 and 173 vertical
profiles, respectively). Temperature and salinity measurements were also extracted to calculate oxygen solubility. The
oxygen data were downloaded from the Argo Global Data Assembly Center web portal accessible through the Coriolis
database (http://www.coriolis.eu.org). Calibration of dissolved oxygen was performed at the deployment using in situ
observations from 0 to 1000 m depth (Winkler titration). The calibration and the deployment strategy are detailed in
(Thierry et al., 2021). The uncertainties of the measurements were estimated at ~ 2-10 µmol kg$^{-1}$ depending on the
sensor (Grégoire et al., 2021).

**2.1.2 Cruise data and seawater measurements**

During the period from October 2018 to March 2019, biogeochemical measurements were made in the context of
PERLE (D'Ortenzio et al., 2021) in the Levantine Basin to describe the preconditioning and dispersion of the
Levantine Intermediate Water (LIW) and to assess its role in structuring the phytoplankton ecosystem. In this study,
we         use         data         from         two         PERLE         cruises:         PERLE-1
(https://campagnes.flotteoceanographique.fr/campagnes/18000848/fr/) on board R/V l'Atalante in October 2018 and
PERLE-2 on board R/V Pourquoi Pas? in February-March 2019 (Conan and Durrieu De Madron, 2019). PERLE-1
covers the period of the preconditioning of LIW formation during which an array of 25 CTD casts was set up. PERLE-
2 covers the water formation period with 29 CTD oxygen casts (Fourrier, 2020). Stations of PERLE cruises are
indicated in Fig. 1. Winkler analyses were performed onboard using photometric endpoint detection to adjust the
SBE43 raw data. After sensor coefficient adjustment, the accuracy of the SBE43 sensor is estimated to be around 2
µmol kg$^{-1}$.
We also use the observations included in the CARIMED database, collected during four cruises that covered the
farthest east and south of the basin: the Meteor M84/3 (Tanhua, 2013), HOTMIX (Arístegui, J., & UTM-CSIC. (2018).
HOTMIX Cruise, RV Sarmiento de Gamboa [Data set]. UTM-CSIC. http://doi.org/10.20351/29SG20140427
SMASH), MEDSEA (Ziveri and Grelaud, 2015), and MSM72 (Hainbucher et al., 2020) cruises conducted during the
period between 2011-2018. The seawater was collected using Niskin bottles from the surface to 4600 m of depth using
an SBE43 oxygen sensor for the oxygen concentrations, followed by the modified Winkler potentiometric method
(Martínez-Pérez et al., 2017). The spatial coverage of the datasets is shown in Fig. 1.

**3 Results**

**3.1 Assessment of the model**

An assessment of the hydrodynamic simulation was performed by Estournel et al. (2021), who showed the capacity
of the model to reproduce the observed hydrology, as well as the surface and intermediate circulations. In Habib et al.
(2023), the results of the biogeochemical model were evaluated in terms of spatial and temporal variabilities of
chlorophyll, dissolved inorganic nutrients, and dissolved oxygen against satellite, cruise, and BGC-Argo float data.
Here, we further assess the dissolved oxygen dynamics by providing supplementary comparisons with cruises, notably
PERLE cruises, and Argo float data, as well as in situ measurements of metabolic rates.




### 3.1.1 Comparison with BGC-Argo float data

Figure 2 represents the temporal evolution of vertical profiles of oxygen from both the model and float observations over the first 500 m, as well as the surface oxygen concentration and oxygen solubility along the float's pathways that were selected for their broad spatio-temporal coverage, each capturing stratification and mixing periods at different locations (Fig. 2a). The observations and the model outputs exhibit the same seasonal variability (Fig. 2 d,e). During summer, both oxygen solubility and surface oxygen concentrations reach their minimum values. Oxygen solubility starts increasing following this period (Fig. 2d-e). Modeled and observed surface oxygen concentrations and solubility show correlation coefficients higher than 0.95 (p-value < 0.05) with a bias lower than 3 µmol $O_2$ kg$^{-1}$. The RMSD values (Root Mean Square Difference) are less than 5 µmol $O_2$ kg$^{-1}$ for both floats and fall within the oxygen uncertainty interval associated with Argo float data.

The general observed features across the water column are respected by the model (Fig. 2b-c) with (i) a subsurface oxygen maximum formation in March/April when the water column restratifies, (ii) a deepening of oxygen maximum until December, followed by (iii) the erosion of the oxygen maximum and homogenization of the surface layer when vertical mixing intensifies, inducing a relatively deep mixed layer. The oxygen maximum in the subsurface layer is underestimated by ~5 µmol kg$^{-1}$ by the model ($\mathtt{RMSD} \simeq \mathtt{8\ \mu mol\ kg}^{-1}$) for the period while its depth is well located. Further deep, the concentration and the localization of the OML are well reproduced with a magnitude of 180 µmol kg$^{-1}$ and depths between 380 and 500 m (Fig. 2b-c). Overall, the simulation reproduces correctly the spatial and temporal variability of the oxygen observed at the surface and in the water column.





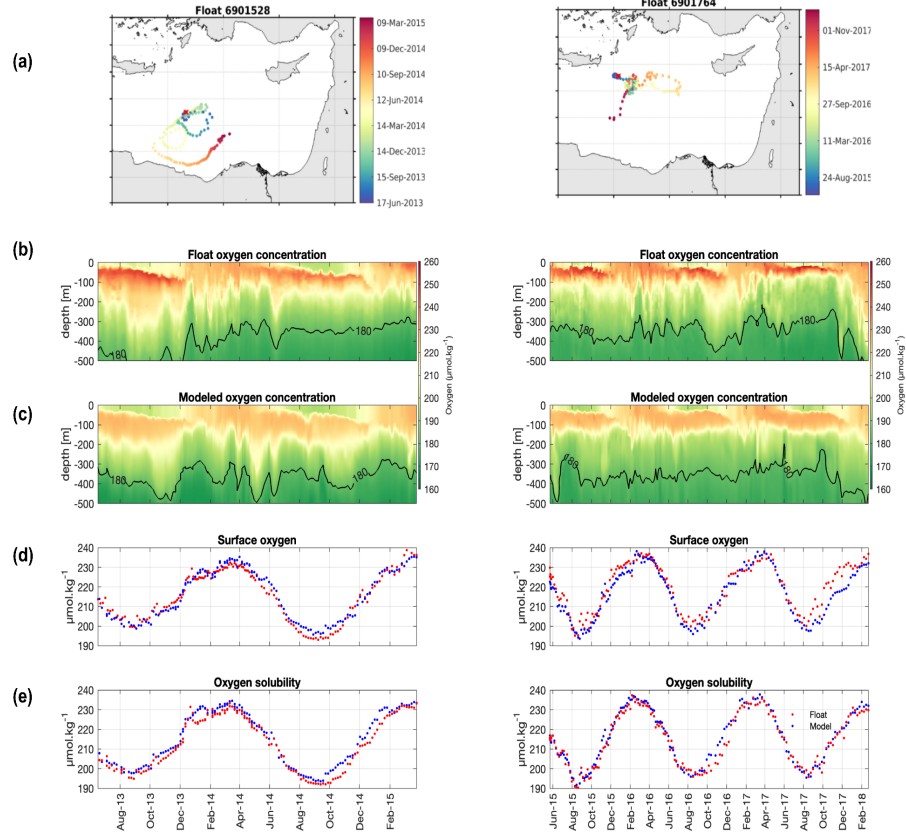

241

**Figure 2: From top to bottom: (a) trajectory of the BGC-Argo floats with deployment position (red cross) and chronology in color; Hovmöller diagrams of oxygen concentration (μmol O₂ kg⁻¹) from (b) float data and (c) model outputs for the first 500 m; (d) surface oxygen concentration in the first 10 m (μmol O₂ kg⁻¹) and (e) oxygen solubility (μmol O₂ kg⁻¹), from the float data (red) and the model (blue).**

**3.1.2 Comparison with cruise data**

Comparisons with data from PERLE-1, PERLE-2, and CARIMED cruises (Fig. S1) show that the model accurately reproduces the magnitude and variability of oxygen concentrations across the different water layers, consistent with observations from the floats comparison. The model and the data from PERLE1, PERLE2, and CARIMED significantly correlate (correlation coefficient higher than 0.95, p-value< 0.05). The model reproduces the intensity of the subsurface maximum reaching 230 μmol O₂ kg⁻¹ during the PERLE 1 cruise in fall (Fig. S1a) contrary to its underestimation noted when comparing the model with the floats (Sect. 3.1.1). These differences between the floats and the cruise data and between the cruises (Fig. S2) could reflect differences in the sampling methods between each campaign of the dataset or mesoscale variability not correctly reproduced by the model.



The highest concentrations (230 µmol O$_2$ kg$^{-1}$) are located at the surface during the winter PERLE 2 cruise and
CARIMED observations collected during the mixing period, for both model and observations (Fig. S1 b and d).
Modeled dissolved oxygen concentrations below 100 m generally stand within the upper range of the observed values,
except for the comparison with PERLE-2 observations, for which the model shows higher values all over the water
column. The overestimation could be attributed to an overestimation of remineralization in this layer or vertical
diffusion.
**3.1.3 Comparison of process rates**
Data on metabolic rates in the eastern Mediterranean Sea are scarce, and most available estimates are derived from
observations made during stratified periods. Comparisons between model results averaged over periods from mid-
June to mid-July and process rates measured near the surface and within the 145 m integrated layer in the core of an
anticyclonic eddy in the eastern Levantine Basin during the BOUM cruise (Christaki et al., 2011) show that the
modeled GPP, CR and NCP are in the upper range of the observational measurements. We also compared our model
averages for May over the Levantine Basin with rates measured during the THRESHOLD cruises (Regaudie-de-Gioux
et al., 2009) in the 5-110 m surface layer, and found values in the range of observations. In addition, model values
integrated over the upper 100 m are consistent with those reported during the MINOS cruise (Moutin and Raimbault,
1996) conducted in the Levantine Basin between May and June. These comparisons can be found in Table 1.
**Table 1: Comparison between averaged modeled and observation rates over the same period and along the same region. a:**
**Lagaria et al., (2011), b: Christaki et al. (2011), c: Regaudie-de-Gioux et al. (2009), (d) Moutin and Raimbault (1996).**

| Process | Campaign | Region | Period | Layer | Observation | Model | Reference |
|---|---|---|---|---|---|---|---|
| **GPP** | BOUM | Core of an anticyclonic eddy in the eastern Levantine | mid-June to mid-July | Near the surface | 0.12 ± 0.90 mmol O$_2$ m$^{-3}$ day$^{-1}$ | 1.36 mmol O$_2$ m$^{-3}$ day$^{-1}$ | a |
| | | | | Integrated 145 m | 28-75 mmol O$_2$ m$^{-2}$ day$^{-1}$ | 68 mmol O$_2$ m$^{-2}$ day$^{-1}$ | b |
| | THRESHOLD | Levantine Basin | May | 5-110 m | 0.16-2.93 mmol O$_2$ m$^{-3}$ day$^{-1}$ | 1-8 mmol O$_2$ m$^{-3}$ day$^{-1}$ | c |
| | MINOS | Levantine Basin | May to June | Near the surface | 0.59 ± 0.16 mmol O$_2$ m$^{-3}$ day$^{-1}$ | 1.08 mmol O$_2$ m$^{-3}$ day$^{-1}$ | d |
| | | | May to June | Integrated 100m | 37.9 ± 4.8 mmol O$_2$ m$^{-2}$ day$^{-1}$ | 37 mmol O$_2$ m$^{-2}$ day$^{-1}$ | d |
| **CR** | BOUM | Core of an anticyclonic eddy in the | | Near the surface | 0.38 ± 0.92 mmol O$_2$ m$^{-3}$ day$^{-1}$ | 1.28 mmol O$_2$ m$^{-3}$ day$^{-1}$ | a |



| | | eastern Levantine | mid-June to mid-July | Integrated 145 m | 39-58 mmol $O_2$ m$^{-2}$ day$^{-1}$ | 68 mmol $O_2$ m$^{-2}$ day$^{-1}$ | b |
|---|---|---|---|---|---|---|---|
| | THRESHOLD | Levantine Basin | May | 5-110 m | 0.1-8.2 mmol $O_2$ m$^{-3}$ day$^{-1}$ | 1-7 mmol $O_2$ m$^{-3}$ day$^{-1}$ | c |
| **NCP** | BOUM | Core of an anticyclonic eddy in the eastern Levantine | mid-June to mid-July | Near the surface | -0.26 ± 0.22 mmol $O_2$ m$^{-3}$ day$^{-1}$ | 0.09 mmol $O_2$ m$^{-3}$ day$^{-1}$ | a |
| | | | | Integrated 145 m | 4 ± 15 mmol $O_2$ m$^{-2}$ day$^{-1}$ | 0.9 mmol $O_2$ m$^{-2}$ day$^{-1}$ | b |
| | THRESHOLD | Levantine Basin | May | 5-110 m | -6.4-8.2 mmol $O_2$ m$^{-3}$ day$^{-1}$ | 0-2 mmol $O_2$ m$^{-3}$ day$^{-1}$ | c |

## 3.2 Seasonal variability

### 3.2.1 Atmospheric forcing and vertical mixing

Figure 3 presents the mean annual cycle of the modeled air-sea heat flux, wind stress, mixed layer (ML) depth, and surface temperature, spatially averaged over the Levantine Sea from December 2013 to December 2020. During fall, the decrease in air temperature leads to significant sea surface heat loss, while intensified northern winds weaken stratification, gradually deepening the mixed layer (Fig. 3a-c). The sea surface temperature drops significantly (Fig. 3d). Heat loss events persist through winter, leading the surface temperature to reach a minimum of approximately 17°C and the mixed layer depth to gradually increase, peaking in January/February. The yearly maximum ML depth averaged spatially over the seven years is 108 ± 11 m (Table S1). In March/April, the sea surface starts gaining heat, and the surface temperature increases (Fig. 3a and 3d). The ML abruptly shallows but still exhibits large variations during early spring, in response to the events of continental cold winds. The frequency of intense wind events decreases in late spring/summer (Fig. 3b). Surface temperature reaches maximum values around 28 °C in August (Fig. 3d), and a thin ML settles until October. In the following, the annual cycle is divided into two successive periods based on the vertical mixing intensity. The first period is a mixing period, from October to March, and the second period is a stratified period, from April to September. The two periods were defined based on a mixed layer depth threshold of 25 m, following the criteria used by D'Ortenzio et al. (2008) and Houpert et al. (2015).

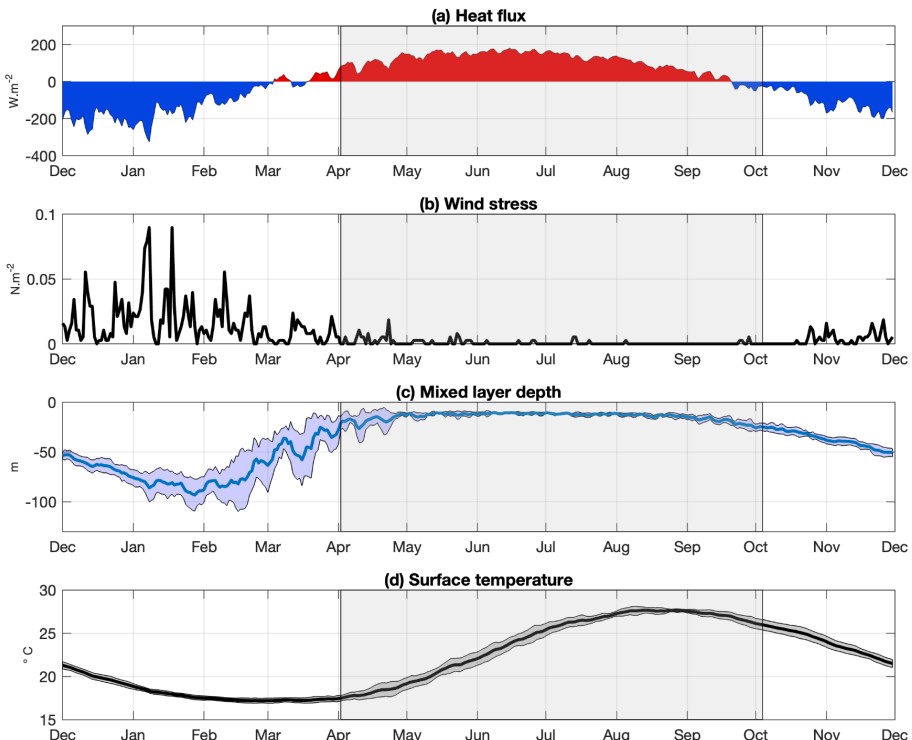

289

**Figure 3: Annual time series of modeled (a) air-sea heat fluxes (W m⁻²), (b) wind stress (N m⁻²), (c) mixed layer depth (m), and (d) surface temperature (°C), averaged over the Levantine Sea and the period 2013-2020. In (c) and (d), the solid line corresponds to the spatial mean, the shaded area to the standard deviation. The grey shaded area represents the stratification period.**

### 3.2.2 Oxygen fluxes

The annual cycle of the 7-year averaged oxygen fluxes and inventory variation are described in Fig. 4 and 5, for the surface and intermediate layers, respectively. The oxygen content in both layers increases during the mixing period and gradually decreases during the stratified period with minimum values in November/December for the surface layer and January for the intermediate layer (Fig. 4a and 5a). The vertical and net horizontal transports show a clear seasonal variation, with the highest values of 50 mmol m⁻² day⁻¹ in winter in the surface layer and 50 mmol m⁻² day⁻¹ and 40 mmol m⁻² day⁻¹ respectively in the intermediate layer (Fig. 4f and 5e). During the mixing period, mostly during intense wind and mixing events (Fig. 3b-c), oxygen is exported from the surface layer towards the intermediate layer (Fig. 4f), and then further down, from the intermediate towards the deep layer (Fig. 5e), with average export rates of 0.56 and 0.46 mol $O_2$ m⁻² month⁻¹, respectively (Fig. S3b,d). During the stratified period, the downward export of $O_2$ towards the intermediate and deeper layers decreases by 75% and 40%, respectively (Fig. S3d). The oxygen horizontal




transport in the surface layer is characterized by a net inflow from the Ionian Sea and an outflow towards the Aegean
Sea (Fig. 4g, S2b), with higher values during the mixing period compared to the stratified period (Ionian Sea: 1.3 vs
0.6 mol $O_2$ $m^{-2}$ $month^{-1}$, and Aegean Sea: 0.9 vs 0.5 mol O2 $m^{-2}$ $month^{-1}$, respectively, Fig. S3b). The horizontal
exchanges in the intermediate layer show a less intense pattern with a stronger net inflow from the Ionian Sea (0.2
mol $O_2$ $m^{-2}$ $month^{-1}$) and a stronger outflow towards the Aegean Sea (-0.4 $O_2$ mol $m^{-2}$ $month^{-1}$, Fig. S3d and 5f) during
the mixing period. The net oxygen flow in the surface layer remains directed from the Ionian and towards the Aegean
across both periods, while the intermediate layer exhibits oxygen outflow toward both the Aegean and Ionian Seas
(96% and 4% of the total horizontal export) during the stratified period, respectively (Fig. S3d).
The model results show that the Levantine Basin ecosystem in the surface layer produces dissolved oxygen from
January to August, at higher rates between February and March, and consumes dissolved oxygen between September
to December (Fig. 4d). Biogeochemical $O_2$ flux (NPC) accounts for a loss of 0.03 mol $m^{-2}$ $month^{-1}$ over the mixing
period (from October to April) and a gain of 0.06 mol $m^{-2}$ $month^{-1}$ during the stratified period (Fig. S3a). The maximum
magnitudes of biological production (> 2 mmol $O_2$ $m^{-2}$ $day^{-1}$) are located near the surface during the periods of winter
mixing and phytoplankton bloom, and then in the subsurface (Fig. 6f). The consumption is maximum in fall between
100 and 200 m depth, and during the mixing period below the mixed layer. During the stratified period, the relatively
thick subsurface oxygen maximum (SOM) layer remains around 70 m depth, above the subsurface maximum
biological production located at around 140 m depth close to the deep chlorophyll maximum (Fig. 6b, 6c, and 6f).
This is in agreement with the findings of Di Biagio et al. (2022). As for the intermediate layer (150-400 m), it shows
a loss of oxygen with values of biogeochemical flux above -10 mmol $m^{-2}$ $day^{-1}$ over the whole year (Fig. 5b).

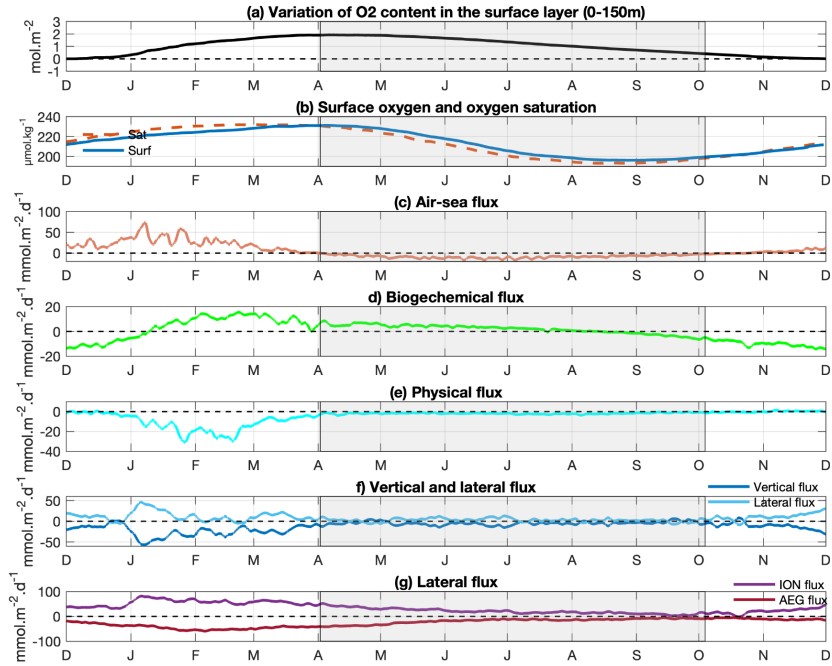






**Figure 4: Oxygen concentration and budget of the 0-150 m layer of the Levantine Basin averaged over the period of study. (a) Variation of the dissolved oxygen inventory (mol m$^{-2}$) relative to initial conditions (Values are normalized to the starting time point), (b) surface oxygen concentration (blue) and oxygen solubility (orange) ($\mu$mol O$_2$ kg$^{-1}$), (c) air-sea flux (positive values correspond to downward fluxes, mmol O$_2$ m$^{-2}$ day$^{-1}$), (d) biogeochemical flux (mmol O$_2$ m$^{-2}$ day$^{-1}$), (e) sum of vertical (through the 150 m depth) and lateral (exchanges with the Ionian and Aegean Seas) transport fluxes (mmol O$_2$ m$^{-2}$ day$^{-1}$), (f) vertical (light blue) and lateral (dark blue) fluxes (mmol O$_2$ m$^{-2}$ day$^{-1}$), (g) lateral fluxes at the boundary with the Ionian (purple) and Aegean (red) Seas (mmol O$_2$ m$^{-2}$ day$^{-1}$). Horizontal transport fluxes are scaled to the area of the Levantine basin for comparison with the other budget terms. The grey shaded area represents the stratification period.**

The air-sea flux displays a seasonal pattern (Fig. 4b). During the October-April mixing period, the Levantine Basin, undersaturated in oxygen compared to the atmosphere, is a sink of atmospheric oxygen. Oxygen solubility has increased (Fig. 4b) with the decrease in surface temperature (Fig. 3d) since September. In parallel, the gradual deepening of the mixed layer favors an increase in the surface oxygen concentration, through the mixing of surface O$_2$ poorer waters with subsurface O$_2$-rich waters (Fig. 6c), which remains lower than the oxygen at saturation level. The air-sea flux, particularly strong in winter when the wind is intense, reaches maximal values around 70 mmol m$^{-2}$ day$^{-1}$ beginning of January (Fig. 4c). The air-sea flux averaged during the mixing period amounts to 0.50 mol O$_2$ m$^{-2}$ month$^{-1}$ (Fig. S3a). At the onset of the stratified period (April - May), surface oxygen concentration reaches 230 $\mu$mol kg$^{-1}$ (Fig. 4b), slightly exceeding saturation levels due to biological oxygen production in the surface layer (Sect. 3.2.4). As a result, the Levantine Basin becomes a source of oxygen for the atmosphere (Fig. 4c). During the rest of the stratified period, the surface oxygen concentration continues to present values higher than the oxygen solubility, leading to continuous outgassing of O$_2$. We estimate a mean net release of 0.26 mol O$_2$ m$^{-2}$ month$^{-1}$ of oxygen to the atmosphere over the whole stratified period (Fig. S3a).





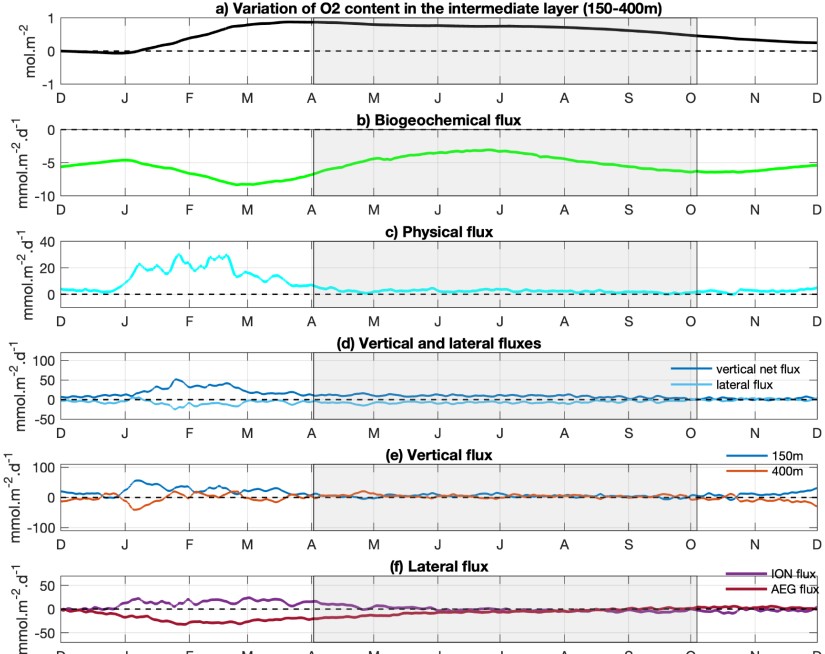

**Figure 5: Mean annual cycle of (a) variation of the dissolved oxygen inventory (mol m$^{-2}$) relative to initial conditions (Values are normalized to the starting time point), and the different oxygen fluxes (mmol m$^{-2}$ day$^{-1}$): (b) biogeochemical flux, (c) total vertical and horizontal transport, (d) vertical (downward) flux (light blue) and lateral flux (dark blue), (e) the vertical fluxes at 150 and 400m and (f) the lateral Ionian (purple) and Aegean (red) fluxes, in the intermediate layer (150-400 m) and averaged over the Levantine Basin. The grey shaded area represents the stratification period.**

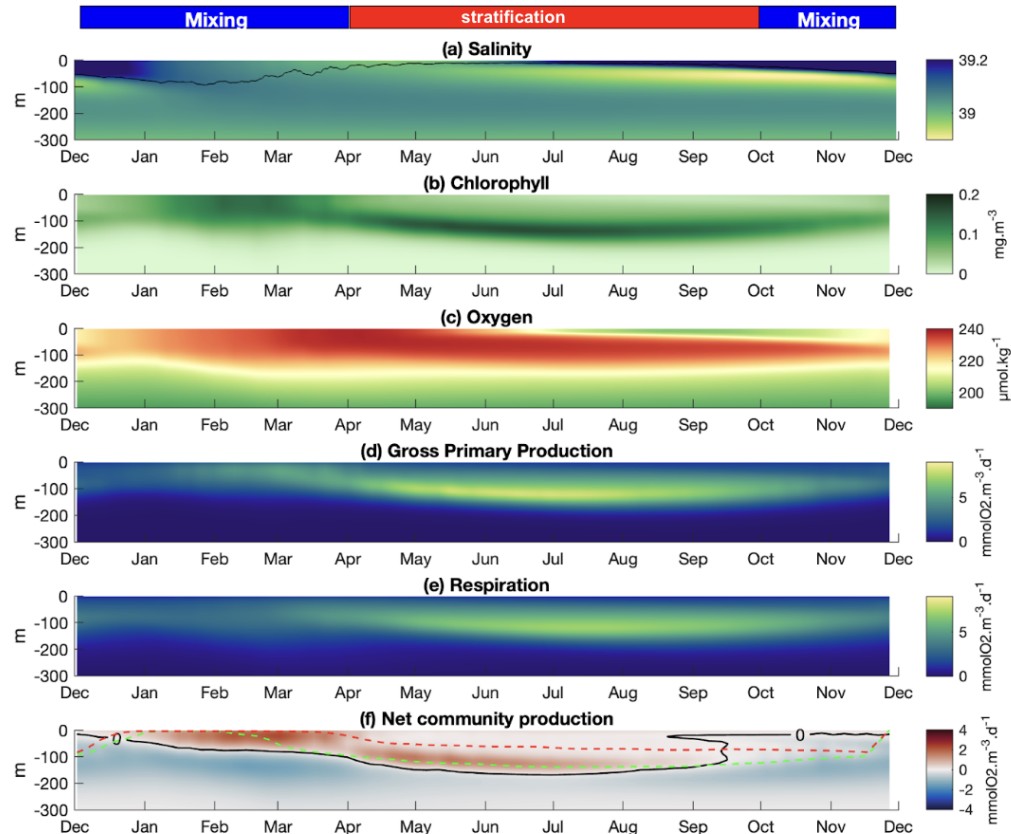

**Figure 6: Hovmöller diagrams of (a) salinity, (b) chlorophyll (mg m⁻³), (c) dissolved oxygen concentration (μmol kg⁻¹), (d) gross primary production (mmol m⁻² day⁻¹), (e) community respiration (mmol m⁻² day⁻¹), and (f) net community production (mmol m⁻² day⁻¹), averaged over the Levantine Basin, from December 2013 to May 2021. The black line in (a) indicates the mixed layer depth.**

## 3.3. Interannual variability

We investigated correlations between processes operating at the annual scale and seasonal fluxes, identifying winter as the most influential period. In the following section, we therefore focus our analysis on winter. For the purposes of this study, the year is defined as running from December to the following December.

### 3.3.1 Atmospheric forcing and vertical mixing

Winter (December-January-February) heat loss exceeded the seasonal mean value of 152 W m⁻² for years 2014-15, 2016-17, 2018-19, and 2019-20 (Fig. 7a, Table S1). In contrast, wind stress does not show consistent interseasonal variability, with peak values around 0.2 N m⁻² occurring each year (Fig. 7b). The ML depth presents interannual variability mostly associated with heat loss fluxes (Fig. 7a and 7c), with higher values than the averaged one of 108



m during the winters 2014-15, 2016-17, and 2018-19 (Table S1). Based on the mean winter heat flux, and mean and
maximum ML, the years were classified as mild and cold winter years: years with both winter heat loss and maximum
ML above the seven-year means, i.e. 2014-15, 2016-17 and 2018-19, are considered cold winter years while, 2013-
14, 2015-16, 2017-18 and 2020-21 are considered mild years.

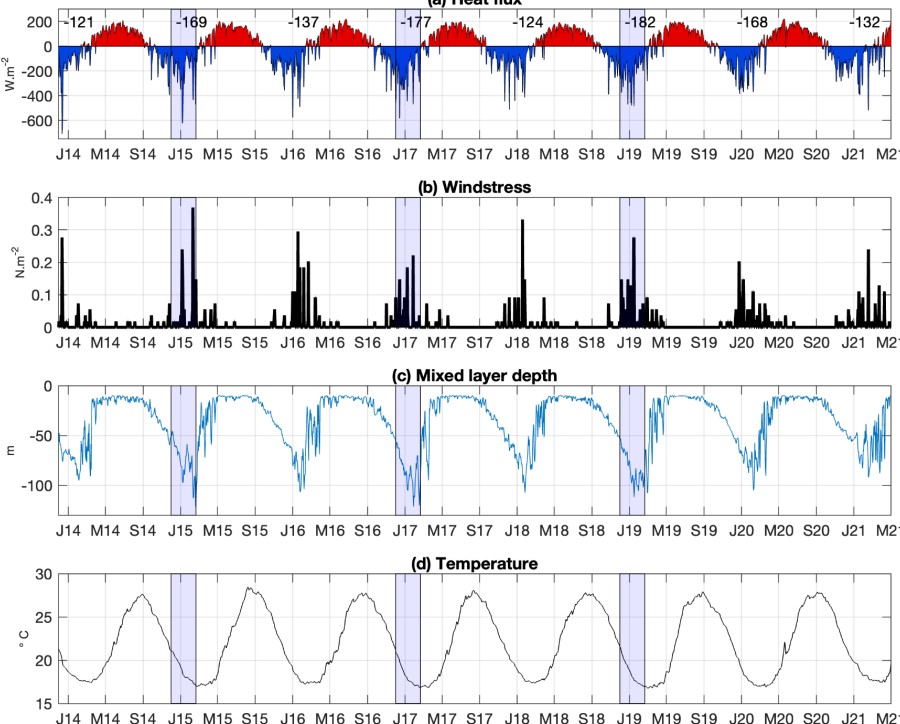


**Figure 7: Time series of modeled (a) air-sea heat fluxes (W m⁻²), (b) wind stress (N m⁻²), and (c) mixed layer**
**depth (m) averaged over the Levantine Basin. The mean winter (December-January-February) heat loss is**
**indicated in (a). The blue shaded area represents the winter of the cold winter years.**
**3.3.2 Dissolved oxygen fluxes in the surface and intermediate layers**
All years display qualitatively similar seasonal oxygen cycles in the surface layer (Fig. 8). However, colder years
(2014–15, 2016–17, 2018–19) exhibit the highest fluxes and greater inventory variations. In winter, oxygen inventory
changes exceed 2 mol $O_2$ m⁻² (Fig. 8a). During phytoplankton blooms in these cold winters, biogeochemical fluxes
surpass 20  mmol m⁻² day⁻¹ (Fig. 8d). Lateral and vertical transports of dissolved oxygen at the surface layer
boundaries show peaks of inflow from the Ionian Sea, outflow toward the Aegean Sea, and downward export to the
intermediate layer, exceeding 100, 75, and 100 mmol m⁻² day⁻¹, respectively (Fig. 8f–g). Air-sea oxygen fluxes also
exceed 150 mmol m⁻² day⁻¹ (Fig. 8c).



In the intermediate layer, cold winter years also present the highest $O_2$ winter fluxes and oxygen inventory variation
compared to mild winter years (Fig. 9). Biogeochemical flux exhibits pronounced negative peaks exceeding 10 mmol
$m^{-2}$ $day^{-1}$ during cold winter years (Fig. 9b). While these negative fluxes are not significantly stronger than in warm
years, the subsequent positive fluxes are markedly enhanced following cold winters (less observed during 2016-17).
Physical fluxes also are marked by a higher magnitude (Fig. 9c-f). As found in the upper layer, higher lateral and
vertical oxygen exchanges occur during the cold winters. Along with the seasonal internal variation, an increasing
trend in the inventory is visible from 2013-14 to 2018-19, followed by a decreasing trend until the end of the study
period (Fig. 9a).
Oxygen fluxes are partly linked to winter heat loss (W-HL). There is a strong correlation between winter oxygen
downward export from the surface layer and mean W-HL ($R = 0.76$, p-value $< 0.05$, Fig. S4). Cold years also show
high NCP (Net Community Production: gross primary production (GPP) minus community respiration (CR)) flux,
with a significant correlation between mean W-HL and annual NCP ($R = 0.91$, p-value $< 0.05$, Fig. S4). When W-HL
drops below 135 W $m^{-2}$, the system shifts from autotrophic to heterotrophic. In the intermediate layer, annual oxygen
consumption remains relatively stable but is still significantly correlated with W-HL ($R = 0.94$, p-value $< 0.05$). Air-
sea oxygen fluxes also show strong correlations with W-HL ($R = 0.92$ annually, $R = 0.93$ in winter, p-value $< 0.05$).
While lateral fluxes increase during cold years, only exchanges with the Aegean Sea show a significant correlation
with W-HL ($R = 0.74$ in the surface layer, $R = 0.82$ in the intermediate layer). The correlation with the Ionian Sea is
weaker and not statistically significant ($R = 0.69$, p-value $< 0.08$).

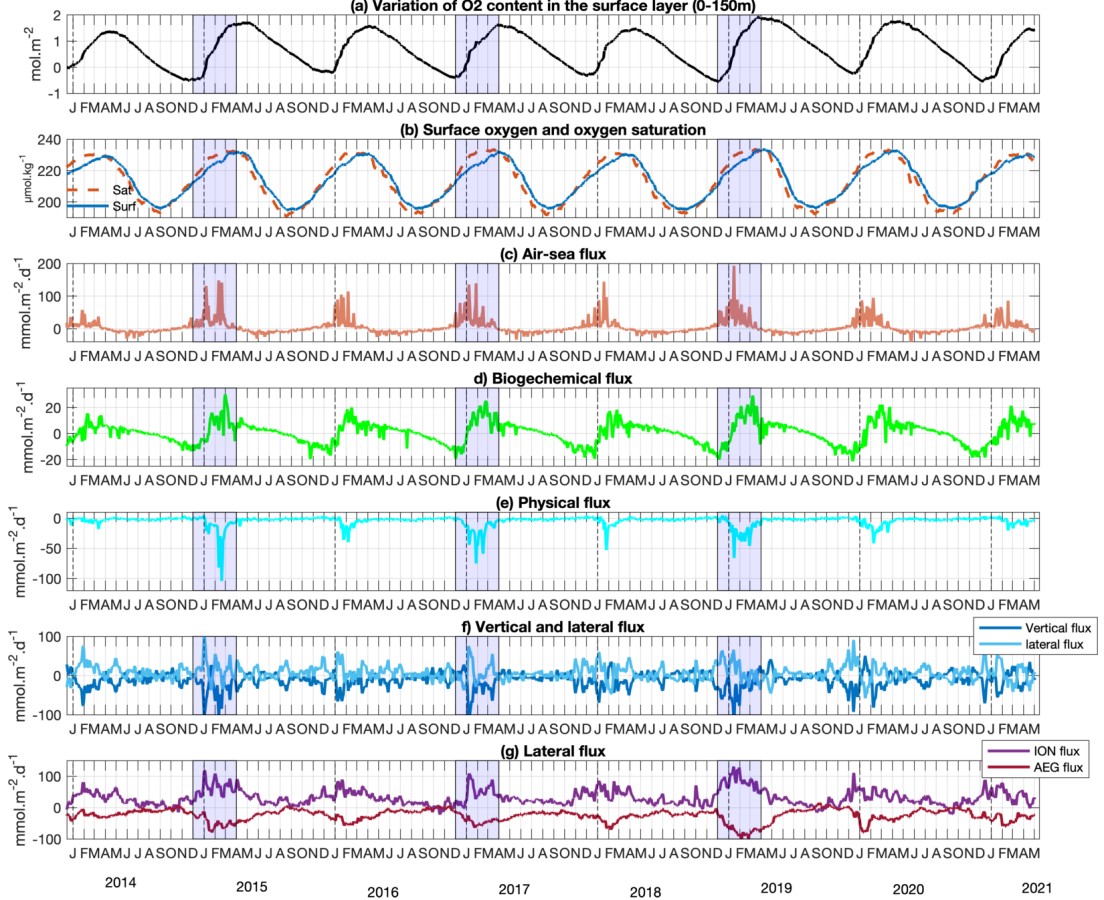

**Figure 8: Time series of oxygen concentration and budget of the 0-150 m layer of the Levantine Basin. (a) Variation of the dissolved oxygen inventory (mol m$^{-2}$) relative to initial conditions (Values are normalized to the starting time point), (b) surface oxygen concentration (blue) and oxygen saturation (orange) (μmol O$_2$ kg$^{-1}$), (c) air to sea flux (mmol O$_2$ m$^{-2}$ day$^{-1}$), (d) biogechemical flux (mmol O$_2$ m$^{-2}$ day$^{-1}$), (e) sum of vertical (through the 150 m depth) and lateral (exchanges with the Ionian and Aegean Seas) transport fluxes (mmol O$_2$ m$^{-2}$ day$^{-1}$), (f) vertical (light blue), lateral (dark blue) flux (mmol O$_2$ m$^{-2}$ day$^{-1}$), (g) detailed lateral flux at the boundary with the Ionian (purple) and Aegean (red) seas (mmol O$_2$ m$^{-2}$ day$^{-1}$). Horizontal transport fluxes are scaled to the area of the Levantine Basin for comparison with the other budget terms. The blue shaded area represents the winter of the cold winter years.**

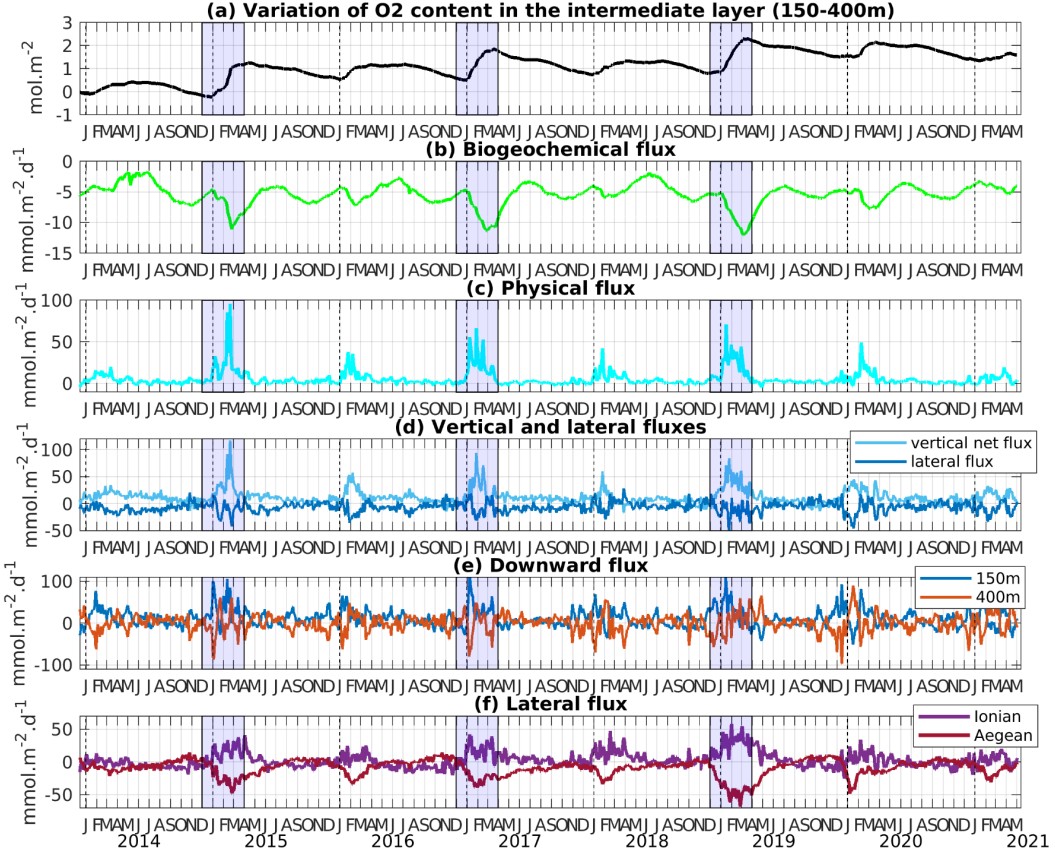

413

**Figure 9: Time series of (a) variation of the dissolved oxygen inventory (mol m$^{-2}$), (b) biogeochemical flux, (c) total vertical and horizontal transport, (d) vertical (light blue) and lateral flux (dark blue), and (g) lateral fluxes from Ionian (purple) and Aegean (red) sea, in the intermediate layer (150-400 m) and averaged over the Levantine Basin. The flux unit is mmol m$^{-2}$ day$^{-1}$. The blue shaded area represents the winter of the cold winter years.**

### 3.4 Annual oxygen budget in the surface and intermediate layers

On an annual basis and on average over the seven studied years (Fig. 10), the surface layer gains oxygen through air-sea exchange. Additionally, oxygen is transported from the Ionian Sea to the Aegean Sea through both surface and intermediate layers. A fraction of this oxygen is exported to intermediate depths, where it is partially consumed before the remainder reaches the Aegean (Fig. 10). The surface ecosystem of the basin appears to be generally productive in oxygen. The net biogeochemical flux is one order of magnitude smaller than transport and air-sea flux, changing sign depending on the considered year, with a strong interannual variability. On average, the intermediate layer loses oxygen due to biogeochemical consumption but gains it through net transport (Fig. 10). The physical supply of oxygen



427    comes from both downward inflow from the surface layer and lateral inflow from the Ionian Sea, while part of the

428    oxygen is exported toward the Aegean Sea (Fig. 10).

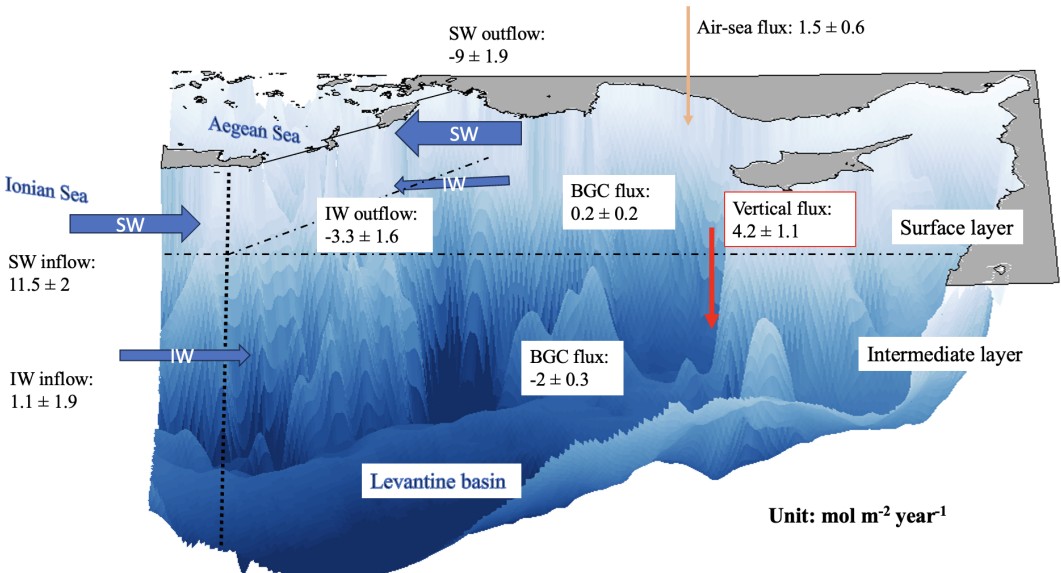

429

**430 Figure 10: Schematic showing the terms of the mean annual oxygen budget (in mol O$_2$ m$^{-2}$ yr$^{-1}$) for the Levantine**

**431 Basin over the period from December 2013 to November 2020. The terms of the budget are estimated for the upper**

**432 layer (surface-150 m), and the intermediate layer (150-400 m). SW: surface layer, IW: intermediate layer. Arrow**

**433 thickness is proportional to the intensity of the flow.**

**434    3.5 Spatial variability of oxygen fluxes in the Levantine Basin**

At the annual scale, the whole Levantine Basin appears as an atmospheric sink for oxygen, except in the coastal area
influenced by the Nile River (Fig. 11a). The highest uptake rates in the offshore region are located in the Rhodes Gyre
area, covering 5% of the surface of the Levantine Basin, and contributing 14% of the annual atmospheric oxygen
intake. Other regions characterized by higher uptake rates are located in the North, in the Antalya Bay and the Cilician
basin. The annual air-sea flux is also spatially mostly controlled by the winter air-sea O$_2$ flux (not shown). The annual
anomalies show that the cold years (2014-15, 2016-17 and 2018-19) absorb more atmospheric oxygen in the whole
sea (Table S2), and especially in the Rhodes Gyre (Fig. S7). In this area of intermediate water formation, the supply
of cold and oxygen-poorer water from the intermediate layer towards the surface during the winter mixing period is
more pronounced than in surrounding areas. Thus, the negative temperature anomaly, as well as the negative oxygen
anomaly, enhanced the undersaturation in this area that shows maximum values varying between 2 and 5 % (with
higher values during cold winter years). In contrast to the entire basin, where oxygen is exported from surface to
intermediate layers, the model indicates that the Rhodes Gyre displayed an inverse pattern, with oxygen transported
upward from intermediate depths to the surface, balanced by a lateral export. This lateral transfer is strong in winter
and takes place notably through the dispersal of LIW by subduction (Estournel et al., 2021). The model results also
show a spatial heterogeneity of the balance between GPP and CR, averaged along the period of study in the surface



layer (surface-150 m) (Fig. 11b). Positive NCP values are found in a central area including the cyclonic Rhodes and
cyclonic gyres (West Cyprus gyres,...), and the coastal areas influenced by rivers. In particular, the Rhodes Gyre
contributes to 41% of the annual biological oxygen production in the surface layer of the whole basin. NCP is negative
in the along-slope circulation, and in the anticyclonic Mersa-Matruh Eddies and Shikmona Eddy. It follows the same
spatial pattern throughout all the years with a more accentuated production in the cyclonic gyres during cold years
compared to the mild years (Fig. S6).

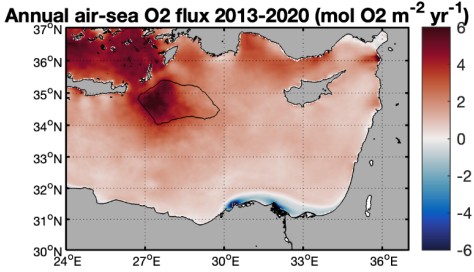 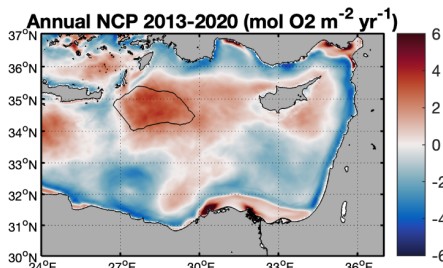


**Figure 11: Modeled annual air-sea oxygen flux and net community production (mol O₂ m⁻² yr⁻¹) in the surface**
**layer (0-150 m) for the period from December 2013 to December 2020. The black line delimits the Rhodes gyre.**
**4 Discussion**
In the present study, we have used a 3D coupled physical-biogeochemical model to investigate the dynamics of oxygen
in the Levantine Basin. The physical and biogeochemical parts of the coupled model were previously validated by
Estournel et al. (2021) and Habib et al. (2023), respectively. Here we have further compared our results on the oxygen
cycle with two types of *in situ* data: high-resolution BGC-Argo data and data from research cruises. The major
limitation highlighted by these comparisons is the representation of the subsurface oxygen maximum layer. The
observed heterogeneity of this layer, with a maximum concentration in the upper part and then a progressive decrease
of its value further deeper, is not well reproduced in the model. The increase of the maximum value during the summer
period, shown in the BGC-Argo data, could be attributed to production or respiration processes, underestimated or
overestimated, respectively, in the model. This discrepancy could also be explained by physical processes with a
misrepresentation of the thickness of the subsurface oxygen maximum layer in the model. In that case, a finer vertical
resolution at those depths or an improvement of the vertical advection scheme, avoiding possible spurious numerical
mixing, as proposed by Garinet et al. (2024), could be tested in future works to reduce the vertical diffusivity. Despite
this limitation, the seasonal variations of oxygen solubility and concentration align with previous observational studies
(Kress and Herut, 2001; Schlitzer et al., 1991) and we used the model to estimate the contribution of various
processes—air-sea flux, physical dynamics, and biogeochemical processes—to oxygen dynamics and the overall
budget.
The model shows a net annual weak biological production of oxygen in the surface layer of the Levantine Basin,
primarily due to the sea's oligotrophic nature, which is more pronounced in the southeastern regions of the Levantine
Basin (D'Ortenzio, 2009; Lavigne et al., 2015). This oligotrophy is attributed to an anti-estuarine circulation
characterized by an eastward inflow of surface nutrient-depleted waters and an outflow of intermediate nutrient-rich



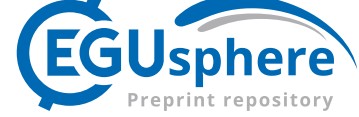

waters resulting from the water formation (Robinson and Golnaraghi, 1993). Interannual variability is observed here,
with the sea being heterotrophic during warm winter years (2014 and 2018 here), as detected by Mayot et al. (2016)
using satellite ocean color data. By considering both surface and intermediate layers, the Levantine Sea appears as a
heterotrophic system in the model results. This aligns with the review by Siokou-Frangou et al. (2010) and the study
by Christaki et al. (2011), depicting a large temporal and spatial heterogeneity in the trophic status of the oligotrophic
Levantine Basin. The planktonic ecosystem in the Levantine Basin is strongly regulated by the basin's heterotrophic
component. Siokou-Frangou et al. (2002) observed an increase in the heterotrophic/autotrophic biomass ratio with
higher values more frequently found in the oligotrophic regions and during the stratified period. This pattern aligns
with the spatial trend reported by Christaki et al. (2002), who depicted a longitudinal increase in this ratio from the
Balearic Sea to the eastern Levantine Sea. Despite this heterotrophic system, highly dynamic mesoscale physical
structures favoring deep vertical mixing, and in particular nutrient upwelling, determine the trophic gradient of the
Levantine Basin (Siokou-Frangou et al., 2010), varying from marked recycled production to new production systems
(Legendre and Rassoulzadegan, 1995). This variability was also detected in nutrient concentrations from different
parts of the Levantine Basin. A shallow nutricline, coupled with an upward nutrient flux, was found in cyclonic
systems, whereas the opposite effect occurred in anticyclonic systems (Salihoğlu et al., 1990). This pattern was also
reflected in the spatial distribution of chlorophyll. Our model results show the high contribution of the Rhodes Gyre
to the annual oxygen biological production in the surface layer of the whole Levantine Basin area (around 41%). In
the intermediate layer, biogeochemical fluxes exhibit little variation between mixing and stratification periods,
especially during cold years, consistent with the findings of Roether and Well (2001) and Klein et al., (2003).
The model indicates that the Levantine Basin absorbs atmospheric oxygen from November to April, while releasing
it during the rest of the year. This is in line with the studies of Schlitzer et al. (1991), Kress and Herut, (2001), and Di
Biagio et al. (2022). The whole Levantine Basin, except the river-influenced areas (Nile river), shows an annual
atmospheric uptake for all studied years ($1.5 \pm 0.6$ mol $O_2$ m$^{-2}$ yr$^{-1}$), with higher values during cold winter years. The
uptake is enhanced in intermediate water formation areas, in particular in the Rhodes Gyre, where the undersaturation
is increased during the winter period, when poorer $O_2$ water masses are mixed with surface waters, in agreement with
what was previously observed and modeled in other water formation areas (Copin-Montégut and Bégovic, 2002;
Coppola et al., 2017, 2018; Di Biagio et al., 2022; Fourrier et al., 2022; Körtzinger et al., 2004, 2008; Ulses et al.,
2021; Wolf et al., 2018). The Rhodes Gyre shows a comparable winter uptake rate ($20.3 \pm 7.4$ mol $O_2$ m$^{-2}$ yr$^{-1}$) as
other water formation areas such as the Labrador Sea and Gulf of Lion (ranging between 11 and 37 mol m$^{-2}$; Copin-
Montégut & Bégovic, 2002; Coppola et al., 2017, 2018; Körtzinger et al., 2008; Ulses et al., 2021; Wolf et al., 2018).
As a matter of comparison, the 7-year averaged oxygen uptake estimated here for the whole Levantine Basin,
characterized by relatively low solubility compared to the rest of the Mediterranean (Mavropoulou et al., 2020, Di
Biagio et al., 2022), represents 64% of the oxygen uptake by the NW Mediterranean deep convection estimated for
the cold year 2012-13 with the same coupled model (Ulses et al., 2021).
Regarding the oxygen vertical transport in the whole Levantine Basin, the weak transfer from the deep layer into the
intermediate layer found in our results is consistent with the general scheme of circulation or oxygen cycle shown in
previous studies (Mavropoulou et al., 2020; Powley et al., 2016; Roether and Schlitzer, 1991; Tanhua et al., 2013)
describing a gradual upwelling of deep water originating from the Adriatic Sea or Aegean Sea. While a downward
export of oxygen from the surface layer to the intermediate layer is simulated at the scale of the whole basin, the
Rhodes Gyre exhibits an opposite pattern, with oxygen being transported upward from the intermediate layers to the



surface. This input in the surface layer is balanced by a lateral export, particularly strong in winter, which takes place
notably through the dispersal by subduction of the newly formed LIW as reported by Malanotte-Rizzoli et al. (2003)
for January 1995 during the POEM cruise.
Our results on lateral oxygen exchanges are also in agreement with previous studies describing the general circulation
in the eastern Mediterranean Sea. Regarding the exchanges with the Aegean Sea, a net outflow of LSW and LIW by
the Asia Minor Current through the Cretan Straits was documented in several observational and modeling studies
(Estournel et al., 2021; Millot and Taupier-Letage, 2005; Theocharis et al., 1993; Velaoras et al., 2014; Zodiatis,
1993). As for the exchanges with the Ionian Sea, the general cyclonic circulation displays in the surface and
intermediate layers an eastward inflow along the Libyan-Egyptian coast (Estournel et al., 2021). South of Crete, the
flux reverses seasonally with an inflow from the Ionian in winter and an outflow in summer (Estournel et al., 2021).
**5 Conclusion**
The study period was marked by contrasted atmospheric and hydrodynamic winter conditions. The confrontation of
the model results with cruise and BGC-Argo float observations shows the capacity of the model to capture the general
seasonal and spatial dissolved oxygen variability and the main oxygen features in the basin. These in situ observations,
particularly from BGC-Argo floats and ship-based measurements, were essential in constraining and validating the
simulations, without which the model outputs would not have reached their current level of reliability. The following
conclusions can be drawn:
- The model results indicate a clear seasonal cycle for the oxygen air-sea flux. During winter, with the decrease
in temperature, the increase in heat losses and mixing events, the surface layer is undersaturated in oxygen
and thus absorbs atmospheric oxygen at the surface. The undersaturation averaged over the whole basin
reaches 2 % during winter. During the stratified period, primary production in the surface layer leads to
oxygen oversaturation and subsequent outgassing, with a maximum oversaturation of 0.6% observed in
summer.
- The Levantine Basin acts as a sink for the atmospheric oxygen at an annual scale, capturing $1.5 \pm 0.6$ mol $O_2$
$m^{-2}$ $yr^{-1}$ of oxygen. Most of the oxygen uptake occurs during winter when it accounts for $10.7 \pm 2.8$ mol $O_2$
$m^{-2}$ $yr^{-1}$. The Rhodes Gyre absorbs atmospheric oxygen at a 2-fold higher rate than the whole Levantine
Basin.
- Our budget shows that the surface layer of the Levantine Basin is a source of dissolved oxygen for the
intermediate depths, with winter vertical export of oxygen influenced by the winter heat loss intensity.
Regarding the exchanges with the surrounding seas, we found that oxygen is laterally transported from the
Ionian Sea into the basin, and from the basin towards the Aegean Sea. The lateral annual oxygen outflow to
the Aegean is strongly enhanced by the heat loss intensity with exports 1.5 and 2.4 times higher during cold
years in the surface and intermediate layer, respectively, compared to mild years.
- The Levantine Basin is found to act as a weak autotrophic ecosystem on an annual level, with a net
community production in the surface layer alternating between auto- and heterotrophic status influenced by
the magnitude of the winter heat loss. In deeper depths, respiration and nitrification resulted in an oxygen



consumption of $2.0 \pm 0.3$ mol $O_2$ m$^{-2}$ yr$^{-1}$. Spatially, the Rhodes Gyre appears to be a major oxygen reservoir
across the basin, contributing 41% of the oxygen production of the whole surface layer.
This work represents a first step in our modeling of the dissolved oxygen dynamics in the Levantine Basin. Further
investigations on the role of the various cyclonic and anticyclonic eddies will be conducted in the future. Besides,
several studies suggest a decadal variability of dissolved oxygen across the whole water column linked to the dense
water formations in the south Adriatic and Aegean seas and to the general eastern Mediterranean circulation, notably
the reversal of the North Ionian Gyre (Ozer et al., 2020, 2022). A time-extended simulation of the coupled model, in
addition to the implementation of a finer vertical resolution at key depths, could contribute to examining this longer
variability in the Levantine Basin and the connections between the sub-basins of the eastern Mediterranean.
**In memoriam**
The authors wish to pay tribute to the memory of Pascal Conan, who passed away on August 5, 2025. He made
insightful contributions and was unwaveringly dedicated to biogeochemical oceanography. We will greatly miss him
both professionally and personally.
**Code availability**
The SYMPHONIE model and the MATLAB codes used to process the model outputs are available from the authors
on request.
**Data availability**
Data used to validate the model are available on different websites specified in the main text of the paper. These data
and the model outputs are also available from the authors on request.
**Author contributions**
CU, CE, and JH conceptualized the study. CE and PM ran the SYMPHONIE model. PM added the budget calculation
to the coupled model. CU and JH calibrated and ran the coupled physical–biogeochemical model. CE validated the
physical model, JH the biogeochemical model. Observational data were provided by PC, MPP, MaF, LC, CWR, DL
and TM. Funding acquisition was done by MiF, CU and CE. JH, CU, and CE wrote the initial version of paper. All
authors contributed to the revision of the paper and approved the submitted version.
**Acknowledgements**
This study is a contribution to the MerMex (Marine Ecosystem Response in the Mediterranean Experiment) project
of the MISTRALS international program. The numerical simulations were performed using the SYMPHONIE model,
developed by the Community Code SIROCCO (https://sirocco.obs-mip.fr/) coordinated by the Research Infrastructure
ILICO (CNRS-IFREMER) dedicated to coastal ocean observations (https://www.ir-ilico.fr/?PagePrincipale, last
access: 16 June 2025), and computed on the cluster of LAERO/OMP and HPC resources from CALMIP grants
(P1331). We acknowledge the scientists and crews of the Flotte océanographique française
(https://www.flotteoceanographique.fr/), who contributed to the cruises carried out in the framework of the PERLE



project. We thank Franck Dumas, chief scientist of the PERLE 1 campaign, for his role in organizing and leading the
cruise. The authors would like to acknowledge the National Council for Scientific Research of Lebanon (CNRS-L),
Campus France, the University of Toulouse, and LEGOS for granting a doctoral fellowship to Joelle Habib. We thank
Marta Álvarez (IEO, La Coruña) and collaborators for making the CARIMED database available to us.
**Financial support**
This research has been supported by the international programme MISTRALS (Marine Ecosystem Response in the
Mediterranean Experiment – MerMex; https://www.odatis-ocean.fr/activites/activites-liees-au-
pole/chantiers/mistrals, last access: 18 August 2025). The numerical simulations were performed with the
SYMPHONIE model developed by the Community Code SIROCCO (https://sirocco.obs-mip.fr/) and coordinated by
the Research Infrastructure ILICO (CNRS–IFREMER; https://www.ir-ilico.fr/?PagePrincipale, last access: 18 August
2025), with computational resources provided by the cluster of LAERO/OMP and CALMIP grants (P1331). The study
also received support from the National Council for Scientific Research of Lebanon (CNRS-L), Campus France, the
University of Toulouse, and LEGOS through a doctoral fellowship granted to Joelle Habib.

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
