# Peer review of "Driving mechanisms of the dissolved oxygen budget in the"

_EGUsphere, 2025_

## Author Comment (AC1)

**Responses to the Reviewers' Comments**

Answers to reviewers' comments are reported point by point. The questions and comments of the reviewers are in blue, the answers in black, and the modifications that we made in the revised manuscript in red.

**Responses to the comments of the anonymous Reviewer 1**

First, we would like to warmly thank Reviewer 1 for their relevant and constructive comments, which helped to improve the manuscript.

*In their manuscript, Habib et al. present a new and well-executed analysis of the mechanisms controlling dissolved oxygen dynamics in the Levantine Sea using a high-resolution coupled physical-biogeochemical model. The manuscript is generally well written, supported by observational validation, and provides valuable insights into the seasonal and interannual variability of dissolved oxygen budget, as well as the key physical-biogeochemical processes involved. I have a few minor comments, outlined below:*

*It would be useful to include a brief discussion on potential sources of uncertainties in the oxygen-heat budget terms (e.g., on gas transfer velocity; L138-140).*

**Reply:** Following the suggestion of reviewer 1, we have added a brief paragraph in the discussion addressing the main sources of uncertainty affecting the oxygen-heat budget terms. We now discuss uncertainties related to the parameterization of the air–sea gas transfer velocity, which depends on wind speed and choice of formulation, as well as uncertainties associated with surface heat fluxes, vertical mixing, and model resolution. We emphasize that these uncertainties do not alter the main conclusions of oxygen variability in the Levantine Basin.

Discussion:

As a matter of comparison, the 7-year averaged oxygen uptake estimated here for the whole Levantine Basin, characterized by relatively low solubility compared to the rest of the Mediterranean (Mavropoulou et al., 2020, Di Biagio et al., 2022), represents 64% of the oxygen uptake by the NW Mediterranean deep convection estimated for the cold year 2012-13 with the same coupled model (Ulses et al., 2021). These estimates are nevertheless subject to methodological uncertainties. In particular, air–sea oxygen fluxes depend on the parameterization of the gas transfer velocity, whose sensitivity to wind speed and formulation can induce uncertainties of the order of 12-16%, as quantified by *Ulses et al.* (2021). Additional uncertainties arise from surface heat flux estimates and the representation of vertical mixing, but these are not expected to modify the relative importance or seasonal phasing of the dominant budget terms (Josey et al., 2013; Large et al., 1994).

References:

Ulses, C., Estournel, C., Fourrier, M., Coppola, L., Kessouri, F., Lefèvre, D., and Marsaleix, P.: Oxygen budget of the north-western Mediterranean deep-convection region, *Biogeosciences*, 18, 937–960, https://doi.org/10.5194/bg-18-937-2021, 2021.

Josey, S. A., Gulev, S., and Yu, L.: Exchanges through the ocean surface, in: *Ocean Circulation and Climate*, 2nd edn., edited by: Siedler, G., Griffies, S. M., Gould, J., and Church, J. A., Academic Press, 115–140, https://doi.org/10.1016/B978-0-12-391851-2.00005-2, 2013.

Large, W. G., McWilliams, J. C., and Doney, S. C.: Oceanic vertical mixing: A review and a model with a nonlocal boundary layer parameterization, *Rev. Geophys.*, 32, 363–403, https://doi.org/10.1029/94RG01872, 1994.

*Since the modelling framework and methods closely follow those used in a previous study by Ulses et al. (2021), I suggest adding a short discussion (and/or in the Introduction) clarifying how the present analysis differs from and builds upon that earlier work. Highlighting the new scientific contributions that emerge from focusing on the Levantine basin will help underline the novelty of the manuscript.*

**Reply:** Following the suggestion of reviewer 1, we have added a short paragraph in the Introduction to clarify how the present study builds upon the framework of Ulses et al. (2021) while addressing distinct scientific questions. While Ulses et al. (2021) focused on basin-scale oxygen dynamics in the western Mediterranean, the present work applies the same modeling framework to the Levantine Basin, a markedly different hydrographic and biogeochemical regime. In particular, this study provides the first basin-wide quantification of the dissolved oxygen budget in the Levantine Basin, with explicit separation of atmospheric, lateral, vertical, and biogeochemical contributions, and highlights the dominant role of the Rhodes Gyre in oxygen uptake and ventilation. This regional focus allows us to document mechanisms specific to the ultra-oligotrophic eastern Mediterranean and to assess how circulation features unique to the Levantine Basin shape oxygen variability.

Introduction:

*The objective of the present work is to quantify the seasonal and interannual variations in the oxygen inventory of the Levantine surface and intermediate water masses over 7 years, detailing the contribution of air-sea oxygen fluxes, biological and physical processes. This analysis is based on 3D coupled hydrodynamic-biogeochemical model outputs covering a period of 7 years, from 2013 to 2020.* *Based on the budget approach developed by Ulses et al. (2021) for the north-western Mediterranean Sea, we investigate the ultra-oligotrophic Levantine Basin and provide a basin-scale quantification of its dissolved oxygen budget, highlighting the role of transport processes and permanent circulation features such as the Rhodes Gyre.*

*Consider linking the results to broader implications for regional biogeochemical research and/or Earth system modelling.*

**Reply:** We agree that placing our results in a broader context strengthens the manuscript. We have therefore expanded the Discussion to highlight the relevance of our findings for regional biogeochemical research and Earth system modelling. Specifically, the quantified oxygen budget and the role of intermediate waters in the Levantine Basin provide insight into how oxygen ventilation and transport in semi-enclosed basins respond to circulation and atmospheric forcing at seasonal to interannual timescales. Because Levantine Intermediate Water is the primary source of Mediterranean Overflow Water, changes in its oxygen content may propagate beyond the Mediterranean and influence the oxygen and biogeochemical properties of intermediate waters in the Northeast Atlantic. Accurately representing these processes is essential for regional biogeochemical models and Earth system models aiming to capture Mediterranean–Atlantic exchanges and their contribution to large-scale oxygen budgets. We have also added a few lines in the conclusion.

Discussion:

*Beyond the Levantine Basin, the processes identified here have broader implications for regional biogeochemical dynamics and Earth system modelling. The quantified oxygen budget of LIW highlights the sensitivity of intermediate-water ventilation to circulation and atmospheric forcing at seasonal to interannual timescales in semi-enclosed basins (e.g. Tanhua et al., 2013; Schneider et al., 2014). Because LIW constitutes the main precursor of MOW, variability in its oxygen content may propagate beyond the Mediterranean and influence the oxygen and biogeochemical properties of intermediate waters in the Northeast Atlantic (Aldama-Campino & Döös, 2020; Stendardo and Gruber, 2012). Accurately representing these processes is therefore essential for regional biogeochemical models and Earth system models aiming to capture Mediterranean–Atlantic exchanges and their contribution to large-scale oxygen budgets.*

Conclusion:

*This work represents a first step in our modeling of the dissolved oxygen dynamics in the Levantine Basin. The quasi-permanent Rhodes Gyre dominates the basin-scale oxygen budget, but transient cyclonic and anticyclonic mesoscale structures are expected to contribute to the oxygen variability outside the gyre at shorter time scales, further investigations on the role of the various cyclonic and anticyclonic eddies will be conducted in the future. While the 7-year study period provides high-resolution insights into oxygen dynamics, it does not cover long-term climate shifts such as the EMT. Several studies suggest a decadal variability of dissolved oxygen across the whole water column linked to the dense water formations in the south Adriatic and Aegean seas and to the general eastern Mediterranean circulation, notably the reversal of the North Ionian Gyre (Ozer et al., 2020, 2022). A time-extended simulation of the coupled model, in addition to the implementation of a finer vertical resolution at key depths, could contribute to examining this longer variability in the Levantine Basin and the connections between the sub-basins of the eastern Mediterranean. Improving the representation of intermediate-water oxygen dynamics in the Mediterranean is also a necessary step toward better quantifying Mediterranean–Atlantic biogeochemical coupling and its sensitivity to future climate-driven changes in ventilation and circulation.*

References:

- Aldama-Campino, A., & Döös, K. (2020). Mediterranean overflow water in the North Atlantic and its multidecadal variability. Tellus A: Dynamic Meteorology and Oceanography, 72(1), 1–10. https://doi.org/10.1080/16000870.2018.1565027
- Stendardo, I., & Gruber, N. (2012). Oxygen trends over five decades in the North Atlantic. Journal of Geophysical Research: Oceans, 117, C11004. https://doi.org/10.1029/2012JC007909

*L90-91: Please add a reference.*

**Reply:** Thank you for this suggestion. A reference has been added at lines 90–91.

*L149-150: specify which product is used.*

**Reply:** The product we used is the Atmospheric Model high resolution forecast (HRES), this was added to the text. *"We used the wind speed and solar radiation from the HRES (Atmospheric Model high resolution forecast) product of ECMWF for the hydrodynamic simulation."*

**Reply:** Thank you for pointing this out. The punctuation in the parentheses at line 451 has been corrected.

*L483-495: The sentences "This aligns ... distribution of chlorophyll" seem tangential and could be shortened for clarity.*

**Reply:** We thank the reviewer for this suggestion. The paragraph at lines 483–495 has been shortened to improve clarity.

The new sentences are as follows:

*"This is consistent with previous studies reporting strong temporal and spatial heterogeneity in the trophic status of the oligotrophic Levantine Basin (Christaki et al., 2011; Siokou-Frangou et al., 2010). The planktonic ecosystem is largely regulated by heterotrophic processes, with higher heterotrophic/autotrophic biomass ratios typically observed in the most oligotrophic regions and during stratified periods (Christaki et al., 2002; Siokou-Frangou et al., 2002). Despite this general heterotrophic character, mesoscale physical structures promoting vertical mixing and nutrient upwelling play a key role in shaping the basin's trophic gradients, particularly within cyclonic systems (Legendre and Rassoulzadegan, 1995; Salihoğlu et al., 1990)."*

*Fig. 11: add a note clarifying the sign convention (i.e., positive/negative = upward/downward fluxes).*

**Reply:** We thank the reviewer for this suggestion. The figure caption has been revised to specify that positive values correspond to a net flux of oxygen into the ocean (uptake), while negative values indicate a net flux from the ocean to the atmosphere. For net community production, positive (negative) values indicate net biological oxygen production (consumption).

*"Figure 11: Modeled annual air-sea oxygen flux and net community production (mol $O_2$ $m^{-2}$ $yr^{-1}$) in the surface layer (0-150 m) for the period from December 2013 to December 2020. The black line delimits the Rhodes gyre. Positive values for the air-sea oxygen flux indicate a net flux of oxygen into the ocean (uptake), while negative values indicate a net flux from the ocean to the atmosphere. Positive NCP values correspond to net biological oxygen production and negative values to consumption."*

---

## Author Comment (AC2)

**Responses to the Reviewers' Comments**

Answers to reviewers' comments are reported point by point. The questions and comments of the reviewers are in blue, the answers in black, and the modifications that we made in the revised manuscript in red.

**Responses to the comments of the anonymous Reviewer 2**

First, we would like to thank Reviewer 2 for their comments, which helped to improve the manuscript.

*General comment*

*The manuscript entitled "Driving mechanisms of the dissolved oxygen budget in the Levantine Sea: a coupled physical-biogeochemical modelling approach" addresses a very interesting and relevant topic. The objectives are undoubtedly important for understanding the variability of the Levantine Basin, a region characterized by high complexity.*

*However, there are several aspects of the work that I find critical and believe should be discussed and reconsidered, as well as some more formal issues.*

*Major technical issues*

*1) The most important aspect of the study is to clearly define the driving mechanisms. I would expect a graphical or quantitative synthesis illustrating the relative contribution of each process to the observed oxygen variability. However, the manuscript reaches the conclusions without discussing the results in sufficient depth in relation to these mechanisms, focusing instead almost exclusively on the oxygen budget. Furthermore, the role of the Rhodes Gyre appears inconsistent throughout the paper. In the abstract, this structure is presented as a key element (which is indeed justified given its importance), but in the conclusion, you state: "Further investigations on the role of the various cyclonic and anticyclonic eddies will be conducted in the future". It would be particularly valuable to assess how these mesoscale structures modulate oxygen variability, not only the Rhodes Gyre, especially since numerous studies have emphasised their importance compared to multi-annual averages, which tend to underestimate total variability.*

*I am not suggesting that your approach is incorrect, but rather that it should account for these processes and their influence on the overall oxygen budget.*

**Reply:** We thank the reviewer for this important and constructive comment. We agree that the manuscript needed clearer emphasis on the physical and biogeochemical processes driving oxygen variability, as well as improved consistency regarding the role of mesoscale structures.

First, to better reflect the actual scope of the study, we have revised the title from *"Driving mechanisms of the dissolved oxygen budget in the Levantine Sea: a coupled physical–biogeochemical modelling approach"* to *"Dissolved oxygen budget in the Levantine Sea: a coupled physical–biogeochemical modelling approach"*. In addition, the objectives of the paper have been reformulated to clearly state the focus and limits of the analyses presented. This change clarifies that the primary objective of the manuscript is the quantification of the oxygen budget and its main

contributors (atmosphere, lateral exchanges, vertical fluxes, biogeochemical fluxes), rather than a comprehensive attribution of all driving mechanisms.

Introduction:

 *[...] In the framework of the PERLE (Pelagic Ecosystem Response to Deep Water Formation in the Levant Experiment) project, the present work aims at quantifying the seasonal and interannual variations in the oxygen inventory of the Levantine surface and intermediate water masses over 7 years, detailing the contribution of air-sea oxygen fluxes, biological and physical processes.*

Regarding mesoscale dynamics, we have clarified the role of the Rhodes Gyre throughout the manuscript. The Rhodes Gyre is a quasi-permanent cyclonic structure and a major site of intermediate water formation, which explains its dominant contribution to oxygen uptake and ventilation compared to transient mesoscale eddies and why we focused on it in this study. This point is now stated more explicitly in the abstract and the conclusion. At the same time, we acknowledge that outside the Rhodes Gyre, mesoscale cyclonic and anticyclonic structures can significantly modulate oxygen variability at shorter time scales. We have revised the conclusions to better articulate that while the present study focuses on basin-scale budgets and the dominant role of a permanent structure such as the Rhodes Gyre, mesoscale eddies represent an important source of variability that deserves dedicated, event-scale analyses beyond the scope of this work.

*Abstract:*

*Spatially, the Rhodes Gyre, a quasi-permanent cyclonic structure and major site of intermediate water formation, emerges as a significant oxygen pump in winter, with annual uptake rates twice as high as the rest of the basin, and shows enhanced biological production in the surface layer during the productive season, contributing to 41% of the net annual oxygen production in the surface layer in the Levantine basin. This study highlights the need for further modeling studies on pluri-annual and multi-decadal scales to explore the interannual variability and evolution of the annual oxygen budget across the entire Eastern Basin, particularly in the context of climate change.*

*Conclusion:*

*This work represents a first step in our modeling of the dissolved oxygen dynamics in the Levantine Basin. The quasi-permanent Rhodes Gyre dominates the basin-scale oxygen budget, but transient cyclonic and anticyclonic mesoscale structures are expected to contribute to the oxygen variability outside the gyre at shorter time scales (Di Biagio et al., 2022), further investigations on the role of the various cyclonic and anticyclonic eddies will be conducted in the future. While the 7-year study period provides high-resolution insights into oxygen dynamics, it does not cover long-term climate shifts such as the EMT. Several studies suggest a decadal variability of dissolved oxygen across the whole water column linked to the dense water formations in the south Adriatic and Aegean seas and to the general eastern Mediterranean circulation, notably the reversal of the North Ionian Gyre (Ozer et al., 2020, 2022). A time-extended simulation of the coupled model, in addition to the implementation of a finer vertical resolution at key depths, could contribute to examining this longer variability in the Levantine Basin and the connections between the sub-basins of the eastern Mediterranean.*

*Another relevant aspect that is not addressed concerns the effect of marine heatwaves. I suggest referring to Figure 3 of the paper "Co-Occurrence of Atmospheric and Oceanic Heatwaves in the Eastern Mediterranean over the Last Four Decades". Similarly, the potential influence of Medicanes*

*(see Menna et al., 2022; Jangir et al., 2023) should at least be briefly discussed. If the manuscript aims to identify and quantify the main mechanisms, these phenomena cannot be entirely overlooked.*

**Reply:** We thank the reviewer for raising the issue of extreme events and their potential influence on oxygen dynamics. Regarding marine heatwaves, we agree that they can affect stratification, air–sea fluxes, and biogeochemical processes, and therefore potentially influence dissolved oxygen variability. While a dedicated analysis of marine heatwaves is beyond the scope of the present study, we have added a discussion in the manuscript to qualitatively address their potential impact on oxygen dynamics in the Levantine Basin, in relation to enhanced stratification, reduced ventilation, and altered air–sea exchanges. Concerning Medicanes, these events are episodic, short-lived, and spatially localized. However, Menna et al. (2023) and Jangir et al. (2024) studies described the influence of such extreme events in the Mediterranean Sea on biogeochemistry, and in particular an increase of phytoplankton concentration and of oxygen solubility. Moreover, in a recent study, Reale et al. (2026) suggests that weather extreme events could induce high $CO_2$ absorption events in winter and significantly contribute to winter air-sea exchanges of carbon. Thus, we acknowledge that Medicanes may locally and temporarily enhance vertical mixing and air–sea exchanges, and could influence the oxygen dynamics. A specific analysis of their influence on oxygen air-sea exchanges and dynamics is also beyond the scope of our study but will need to be considered in future works.

**Discussion**

*[...] In addition to seasonal forcing, extreme atmospheric events may episodically modulate air–sea oxygen exchanges in the Levantine Basin. In particular, marine heatwaves, whose frequency and intensity have increased in the Eastern Mediterranean in recent decades (Aboelkhair et al., 2023; Darmaraki et al., 2024; Reale et al., 2026), can enhance upper-ocean stratification, reduce vertical ventilation, and decrease oxygen solubility through surface warming, potentially leading to transient oxygen anomalies in the upper and intermediate layers (Keeling et al., 2010; Schmidtko et al., 2017). While a dedicated analysis of marine heatwaves is beyond the scope of the present study, their effects may contribute to short-term departures from the mean seasonal oxygen cycle. Moreover, Medicanes are short-lived and spatially localized extreme events, could also impact biological dynamics and air-sea exchanges on their passage (Menna et al., 2023; Jangir et al., 2024; Reale et al., 2026), and their integrated contribution to basin-scale and annual oxygen budgets will need to be assessed in future works.*

References:

Aboelkhair, H., Mohamed, B., Morsy, M., & Nagy, H. (2023). Co-occurrence of atmospheric and oceanic heatwaves in the Eastern Mediterranean over the last four decades. *Remote Sensing*, **15**, 1841. https://doi.org/10.3390/rs15071841

Darmaraki, S., Denaxa, D., Theodorou, I., Livanou, E., Rigatou, D., Raitsos, E. D., Stavrakidis-Zachou, O., Dimarchopoulou, D., Bonino, G., McAdam, R., Organelli, E., Pitsouni, A., & Parasyris, A. (2024). Marine heatwaves in the Mediterranean Sea: A literature review. *Mediterranean Marine Science*, **25**(3), 586–620. https://doi.org/10.12681/mms.38392

Reale, M., F. Giordano, V. Di Biagio, G. Cossarini, S. Salon (2026). Co-occurrence of atmospheric and oceanic heatwaves in the Eastern Mediterranean over the last four decades. *Journal of Geophysical Research: Atmospheres*, **130**, e2025JD044310. https://doi.org/10.1029/2025JD044310

Keeling, R. F., Körtzinger, A., & Gruber, N. (2010). Ocean deoxygenation in a warming world. *Annual Review of Marine Science*, **2**, 199–229. https://doi.org/10.1146/annurev.marine.010908.163855

Jangir, B., Mishra, A. K., & Strobach, E. (2024). The interplay between medicanes and the Mediterranean Sea in the presence of sea surface temperature anomalies. *Atmospheric Research*, **310**, 107625. https://doi.org/10.1016/j.atmosres.2024.107625

Menna, M., Martellucci, R., Reale, M., Cossarini, G., Salon, S., Notarstefano, G., et al. (2023). A case study of impacts of an extreme weather system on the Mediterranean Sea circulation features: Medicane Apollo (2021). *Scientific Reports*, 13(1), 3870. https://doi.org/10.1038/s41598-023-29942-w

Schmidtko, S., Stramma, L., & Visbeck, M. (2017). Decline in global oceanic oxygen content during the past five decades. *Nature*, **542**, 335–339. https://doi.org/10.1038/nature21399

*2) In Estournel et al. (2021), the authors clearly state: "It should also be kept in mind that our simulation is also related to a period characterised by cyclonic circulation in the Ionian Sea." Later in your discussion section, you write: "Our results on lateral oxygen exchanges are also in agreement with previous studies describing the general circulation in the eastern Mediterranean Sea." However, it is not entirely clear whether these previous studies describe the classical Levantine Basin circulation or rather conditions associated with an EMT-type regime. For instance, I am not convinced that Zodiatis et al. (1993) depicts the same circulation pattern between the Cretan Sea and Levantine Basin described in your work.*

**Reply:** You are absolutely right: this statement is too simplistic, and indeed Zodiatis' results are also more complex in terms of circulation at the various straits closing the Cretan Sea to the west and the east. We therefore propose to restrict these results to the cyclonic phase of the BIOS. To answer this question, we suggest changing this sentence to:

*Our results on lateral oxygen exchanges are also in agreement with previous studies describing the general circulation in the eastern Mediterranean Sea under BIOS cyclonic phases. Regarding the exchanges with the Aegean Sea, a net outflow of LSW and LIW by the Asia Minor Current through the Cretan Straits was documented in several observational and modeling studies (Estournel et al., 2021; Millot and Taupier-Letage, 2005; Theocharis et al., 1993; Velaoras et al., 2014, Zodiatis, 1993). As for the exchanges with the Ionian Sea, the general cyclonic circulation displays in the surface and intermediate layers an eastward inflow along the Libyan-Egyptian coast (Estournel et al., 2021). South of Crete, the flux reverses seasonally with an inflow from the Ionian in winter and an outflow in summer (Estournel et al., 2021). During anticyclonic phases of the BIOS, significant changes in oxygen circulation are to be expected and deserve further investigation.*

*Since your model are forced by that physics what extent does the biogeochemical component depend on them? This is a crucial issue, as the eastern Mediterranean is an extremely complex area where both mesoscale and basin-scale variability strongly influence the oxygen dynamics.*

**Reply:** You are absolutely right. Different results can be expected if other models are used. However, it can be said that the physical model and the biogeochemical model have been extensively compared with observations in Estournel et al. (2021), Habib et al. (2023), and in this paper, which gives some

confidence in the results. Despite this, there are of course many uncertainties that are difficult to quantify without performing further simulations. Performing ensemble and/or multi-model simulations is an interesting avenue to explore in this regard. We suggest mentioning this in the discussion following the previous point. Indeed, considering only one phase of the BIOS is also reductive.

*Discussion:*

*Our budget of oxygen is subject to sources of uncertainties linked to the physical and biogeochemical models used in this study. One approach to overcome single-model uncertainties and limitations could be to perform a multi-models approach. An alternative approach would consist in estimating oxygen budgets using observational syntheses; however, the sparse spatial coverage of in situ data currently limits the closure of basin-scale oxygen budgets based solely on observations. Finally, a combined approach such as the one developed by Di Bagio et al. (2023) using the Copernicus Marine MediterraneanSea biogeochemical reanalysis corrected with Argo float data could also be considered to estimate a budget in the area in future works.* Despite these limitations, we found that the seasonal variations of oxygen solubility and concentration align with previous observational studies (Kress and Herut, 2001; Schlitzer et al., 1991) and *for this first assessment* of the contribution of various processes—air-sea flux, physical dynamics, and biogeochemical processes—to oxygen budget *we chose an online and strictly closed budget approach. Trinh et al. (2024) demonstrated that this approach can yield substantially higher accuracy than offline calculations, especially in the quantification of lateral fluxes.*

 *[...]*

 *[...]Our results on lateral oxygen exchanges are also in agreement with previous studies describing the general circulation in the eastern Mediterranean Sea under BIOS cyclonic phases.* Regarding the exchanges with the Aegean Sea, a net outflow of LSW and LIW by the Asia Minor Current through the Cretan Straits was documented in several observational and modeling studies (Estournel et al., 2021; Millot and Taupier-Letage, 2005; Theocharis et al., 1993; Velaoras et al., 2014; Zodiatis, 1993). As for the exchanges with the Ionian Sea, the general cyclonic circulation displays in the surface and intermediate layers an eastward inflow along the Libyan-Egyptian coast (Estournel et al., 2021). South of Crete, the flux reverses seasonally with an inflow from the Ionian in winter and an outflow in summer (Estournel et al., 2021). *During anticyclonic phases of the BIOS, significant changes in oxygen circulation are to be expected and deserve further investigation.*

*The Levantine Basin experiences thermohaline oscillations driven by the Ionian circulation (as you mention in your conclusions and in to the introduction citing Marvopoulou et al., 2020 and Ozer et al., 2022). During anticyclonic phases in the Ionian, the inflow of Atlantic water is reduced, leading to increased salinity in the Levantine basin. This affects not only surface layers but also intermediate and deep waters, both in the Levantine Basin (as discussed by Ozer et al. 2022) and in the Adriatic (Martellucci et al., 2024; Civitarese et al., 2023), showing opposite behavior between the two basins. This inverse relationship is also evident in Mavropoulou et al. (2020, Fig. 9). Furthermore, Liu et al. (2021) (Fig. 4a, Drivers of the decadal variability of the North Ionian Gyre upper layer circulation during 1910–2010) clearly shows how oxygen minima and maxima alternate following the variability of the NIG at longer time scale.*
*Between 2013 and 2020, three circulation inversions occurred, influencing both the surface water inflow and the distribution of the Oxygen Minimum Layer (OML) associated with Transitional Mediterranean Water. For instance, Manca et al. (2004) and earlier studies report OML depth variations ranging from 500 to 1700 m,: 1987: OML at 1700 m (Souvermezoglou et al., 1992) 1995:*

*OML at 1100 m (Klein et al., 1999) 1999: OML at 650 m (Manca et al., 2003) 2011: OML at 1000 m (Cardin et al., 2015). In the text, you say that it is located between 600 and 1200, referring to Tanhua et al. (2013). It doesn't seem to me that Tanhua et al., 2013 says the OML is located between 600 and 1200, "In the eastern basin, the OML core lies in the depth range of 500–700 m, well below the layer of maximum S occupied by the LIW"*

**Reply:** We thank the reviewer for pointing this out. The reference to Tanhua et al. (2013) was intended to support the characterization of oxygen concentrations within the oxygen minimum layer, not its vertical extent (Fig. 6 in Tanhua et al.). The depth range of 600–1200 m corresponds instead to the broader oxygen minimum zone described by Mavropoulou et al. (2020). We agree that this distinction was not sufficiently clear in the original text therefore we will explicitly separate these two aspects: Tanhua et al. (2013) is now cited for the oxygen minimum characteristics, while Cardin et al. (2015) and Mavropoulou et al. (2020) are cited for the depth range of the oxygen minimum zone. In addition, we have explicitly acknowledged the role of large-scale circulation variability by adding a sentence on the potential influence of the Adriatic–Ionian Bimodal Oscillating System (BiOS) on oxygen variability. In particular, we now note that changes in Ionian circulation can modulate the thermohaline and biogeochemical properties of water masses entering the Levantine Basin, thereby influencing LIW and oxygen distributions (Ozer et al., 2022).

We have updated the line as follow:

*An Oxygen Minimum Layer (OML) is located between 600 and 1200 m (Cardin et al., 2015; Mavropoulou et al., 2020) with a minimum concentration of 170/180 µmol kg⁻¹ (Tanhua et al., 2013).*

*In light of these considerations, the statement at lines 170–171 ("In this study, we will be focusing on the first two layers, as changes at greater depths are very slow over the 8-year period and barely detectable") should be reconsidered. Variability at intermediate depths may significantly affect the driving mechanisms of oxygen dynamics.*

*Additionally, the sentence at line 70 ("The Levantine Basin shows spatial changes in oxygen content occurring at short, annual, and decadal time scales") appears inconsistent with the above assumption.*

**Reply:** We thank the reviewer for this comment and agree that the wording at lines 170–171 was too strong and could be misleading. While our results show that oxygen changes at the deep layer are relatively small over the 8-year period considered, variability at these depths remains dynamically important. The sentence has therefore been removed to avoid implying that deep layer variability is negligible, and we will then keep "The Levantine Basin shows spatial changes in oxygen content occurring at various time scales", as it refers to the general behavior of oxygen variability in the Levantine Basin across a range of time scales that was documented in previous studies and not to the specific focus of the present analysis.

*In this study, we will be focusing on the first two layers where changes occur generally more rapidly; as changes at greater depths are very slow over the 8-year period and barely detectable.*

*3) Another aspect that needs to be clarified concerns the statement:*

*"we found that oxygen is laterally transported from the Ionian Sea into the basin, and from the basin towards the Aegean Sea."*

*This pattern appears to be strongly influenced by what is shown in Figure 15 of Estournel et al. (2021), where water enters the Aegean through the Kasos–Karpathos straits, exits through the Kythira and Antikythira passages, and, according to your schematic, re-enters the Levantine Basin. This circulation scheme seems rather unusual: it would imply that water does not flow outward from the Levantine Basin but instead recirculates continuously through its northern part (see Malanotte-Rizzoli et al., 1999).*

*It is well established that the Levantine Intermediate Water (LIW) flows east-to-west (as you correctly mention at line 50). In terms of oxygen dynamics, this reflects a net wintertime atmospheric uptake associated with the formation of intermediate water that subsequently exits the Levantine Basin (zonal thermohaline circulation). In your conclusion, however, the oxygen transport is directed toward the Levantine Basin, which needs to be further explained and justified.*
*For reference, Taillandier et al. (2022), a key experimental study that clearly describes the circulation pattern during part of your study period, reported a particular circulation between 2018 and 2019, when most of the LIW was formed in the Cretan Basin. To what extent does this anomalous LIW formation affect your diagnosed drivers and the oxygen budget?*

**Reply:** We thank the reviewer for this detailed comment and for raising the issue of consistency with the classical LIW circulation scheme. We would like to clarify that the diagnosed lateral oxygen transport from the Ionian Sea into the Levantine Basin at intermediate depths does not imply an eastward transport of LIW. In the model, lateral oxygen fluxes represent the transport of oxygen carried by different water masses and circulation components, including exchanges through the Cretan straits and recirculated or modified intermediate waters originating from the Ionian Sea. When compared to the vertical oxygen flux from the surface layer to intermediate depths associated with winter ventilation and biological production, the lateral oxygen input from the Ionian Sea at intermediate depths is a secondary contribution to the overall oxygen budget (about 3–4 times weaker). The dominant source of oxygen for intermediate waters in the Levantine Basin therefore remains vertical transfer from the surface layer, particularly during winter mixing and convection. These inflows coexist with the well-established westward export of LIW, which is correctly represented in the model and described in the manuscript. Oxygen entering the Levantine Basin at intermediate depths is therefore associated with water masses distinct from newly formed LIW and does not contradict the zonal thermohaline circulation of the eastern Mediterranean. The statement that oxygen is laterally transported from the Ionian Sea into the Levantine Basin and exported toward the Aegean Sea refers to net, time-averaged lateral oxygen fluxes diagnosed over the full 2013–2020 period. These fluxes result from the superposition of seasonal and interannual circulation variability and should not be interpreted as a schematic of the mean circulation pathways or as evidence of a closed recirculation within the northern Levantine Basin. We acknowledge that episodic circulation regimes, such as those described by Taillandier et al. (2022) for 2018–2019, can locally and temporarily modify oxygen transport pathways. Such circulation inversions are not explicitly resolved in the present model configuration; however, their contribution to the diagnosed oxygen budget remains limited when averaged over the 7-year study period. A detailed investigation of regime-dependent circulation reversals and their impact on oxygen dynamics lies beyond the scope of the present study. The manuscript has been revised to clarify this distinction:

*" [...] oxygen is laterally transported  into the Levantine Basin by intermediate waters originating from the Ionian Sea mainly through the surface layer and, to a lesser extent, through the recirculation of modified intermediate waters"*

**4)** *In the conclusions, the authors state that a longer simulation period and a larger domain would be desirable. This raises the question of why the Levantine Basin was treated separately, instead of analyzing the same spatial domain as Estournel et al. (2021). The Cretan Passage is a wide and dynamically active area where mesoscale structures strongly influence water transport, and excluding it may limit the representativeness of the results.*

**Reply:** We would like to clarify a potential misunderstanding regarding the spatial extent of our study. Our simulation domain and budget calculations explicitly include the Cretan Passage. As shown in Figure 1, the western boundary of our budget box (delimited by the black lines) is located at a longitude that encompasses the entire Cretan Passage. This ensures that the lateral transport calculated in our budget (Section 2.1.3) accounts for the significant water mass exchanges and mesoscale structures between the Ionian and Levantine basins. The mention of a larger domain in our conclusion was not meant to imply the inclusion of the Cretan Passage, but rather to suggest the benefit of extending future research to a basin-wide Mediterranean scale (as in Estournel et al., 2021). Such an expansion would allow for a more comprehensive assessment of long-distance connectivity with the Western Mediterranean and the Adriatic Sea. We acknowledge that a slight misalignment of the boundary lines in Figure 10 compared to Figure 1 may have caused this confusion. We have corrected the figures to ensure perfect consistency. We thank the reviewer for pointing out this ambiguity.

[Figure]

**Figure 10: Schematic showing the terms of the mean annual oxygen budget (in mol $O_2$ $m^{-2}$ $yr^{-1}$) for the Levantine Basin over the period from December 2013 to November 2020. The terms of the budget are estimated for the upper layer (surface-150 m), and the intermediate layer (150-400 m). SW: surface layer, IW: intermediate layer. Arrow thickness is proportional to the intensity of the flow.**

*Another point that deserves clarification is the choice of the simulation period. Why was only a 7-year period analyzed? Such a short time span may not adequately capture the full range of variability associated with the oxygen budget, which requires a longer record to be properly understood, for*

*instance, including the EMT and post-EMT periods. These are mentioned in the text but not further discussed.*

**Reply**: We thank the reviewer for this insightful comment regarding the simulation period. We agree that a multi-decadal perspective is essential for fully capturing the long-term variability of the Mediterranean oxygen budget. However, the choice of the 2013–2020 period was guided by several technical and scientific considerations:

1. **Availability of Physical Forcing:** The biogeochemical model is coupled with a high-resolution physical configuration (SYMPHONIE). The hydrodynamic simulation was performed without observation assimilation.
2. **Initialization and Data Density:** The year 2011 was chosen as the starting point because it offered a high density of in situ data, which was crucial for establishing a realistic initial state for the biogeochemical variables.
3. **Spin-up Period:** To ensure the model reached a stable internal equilibrium, the first two years of the simulation (2011–2012) were treated as a spin-up period. Consequently, the budget analysis focuses on the 2013–2020 period to ensure data quality and model stability.
4. **Model Validation:** This period (2013–2020) coincides with the maximum availability of BGC-Argo float data and oceanographic campaigns in the Levantine Basin, allowing for the rigorous validation presented in the paper.

Regarding the EMT: while this major event is fundamental to the basin's history, they occurred outside the temporal window of our high-resolution physical forcing. Discussing them in detail would have been speculative without a corresponding multi-decadal simulation. We have updated Section 2.1 to justify the choice of this period better and have added a statement in the Conclusion noting that extending this work to a decadal scale, including periods like the EMT, is a priority for our future research.

*section 2.1.2*

*The first two years (2011–2012) for the biogeochemical model were dedicated to model spin-up to ensure biogeochemical stability and were not considered in the analysis, while the 2013–2020 period was used for the budget analysis. This period was selected based on the availability of consistent physical forcing and the density of in situ observations for model initialization and validation.*

*Conclusion*

*This work represents a first step in our modeling of the dissolved oxygen dynamics in the Levantine Basin. The quasi-permanent Rhodes Gyre dominates the basin-scale oxygen budget, but transient cyclonic and anticyclonic mesoscale structures are expected to contribute to the oxygen variability outside the gyre at shorter time scales, further investigations on the role of the various cyclonic and anticyclonic eddies will be conducted in the future. While the 7-year study period provides high-resolution insights into oxygen dynamics, it does not cover long-term climate shifts such as the EMT. Several studies suggest a decadal variability of dissolved oxygen across the whole water column linked to the dense water formations in the south Adriatic and Aegean seas and to the general eastern Mediterranean circulation, notably the reversal of the North Ionian Gyre (Ozer et al., 2020, 2022). A time-extended simulation of the coupled model, in addition to the implementation of a*

*finer vertical resolution at key depths, could contribute to examining this longer variability in the Levantine Basin and the connections between the sub-basins of the eastern Mediterranean.*

*5) As a suggestion, since the paper aims to explain the drivers and the oxygen budget, it would be valuable to complement the modeling results with additional biogeochemical datasets. For example, the free Copernicus biogeochemical reanalysis and observational data (including BGC-Argo) could be used to estimate the oxygen budget from independent sources. Both CMEMS and Argo data are publicly available, and several field campaigns have already provided supporting measurements. In particular, Argo oxygen profiles could be used not only for model validation but also to estimate the budget directly, possibly through binning or averaging approaches. During your study period, approximately 13000 Argo oxygen profiles are available in the area of interest. Ship-based data could also be considered, for example, as highlighted in the final conclusions of D'Ortenzio et al. (2021): "An unprecedented BGC-Argo observation system was implemented in the Levantine area of the Mediterranean Sea in 2018–2019. It was supported by an equivalent and concomitant ship-based effort (three seasonal surveys from May 2018 to March 2019) to elucidate the impact of physical forcing on the biogeochemical dynamics of the basin." If another author were to reproduce the oxygen budget using the Copernicus biogeochemical model or observational data following your approach, the results would likely differ, affecting the interpretation of the underlying mechanisms. Incorporating multiple datasets may require additional effort, but it would strengthen your estimates and make the analysis more consistent with the study's stated objectives and title.*

**Reply:** We thank the reviewer for this constructive suggestion. We agree that combining multiple independent datasets, such as Copernicus biogeochemical reanalyses and observational products, can provide complementary perspectives on oxygen dynamics and budget estimates. The objective of the present study, however, is to quantify the dissolved oxygen budget using a single, internally consistent coupled physical–biogeochemical modeling framework that was extensively validated, while using in situ observations (BGC-Argo floats and ship-based measurements) primarily for model evaluation. Estimating a full oxygen budget directly from observations remains challenging due to the sparse and uneven spatial and temporal sampling, as well as the difficulty of closing budget terms (e.g., lateral and vertical transports) from observational data alone. Regarding BGC-Argo data availability, while a large number of oxygen profiles exist in the broader Eastern Mediterranean, the effective number of profiles within the Levantine Basin and during the study period is substantially lower than the total number cited, and their spatial distribution does not allow a robust basin-scale budget reconstruction using binning or averaging approaches. To address the reviewer's point, we have added a sentence in the Discussion acknowledging that multi-dataset and multi-model approaches, including Copernicus reanalyses and expanded observational syntheses, represent a valuable complementary methodology that could be explored in future studies to further assess uncertainties in basin-scale oxygen budgets.

*Discussion:*

*Our budget of oxygen is subject to sources of uncertainties linked to the physical and biogeochemical models used in this study. One approach to overcome single-model uncertainties and limitations could be to perform a*

*multi-models approach. An alternative approach would consist in estimating oxygen budgets using observational syntheses; however, the sparse spatial coverage of in situ data currently limits the closure of basin-scale oxygen budgets based solely on observations. Finally, a combined approach such as the one developed by Di Bagio et al. (2023) using the Copernicus Marine MediterraneanSea biogeochemical reanalysis corrected with Argo float data could also be considered to estimate a budget in the area in future works.* Despite these limitations, we found that the seasonal variations of oxygen solubility and concentration align with previous observational studies (Kress and Herut, 2001; Schlitzer et al., 1991) and *for this first assessment* of the contribution of various processes—air-sea flux, physical dynamics, and biogeochemical processes—to oxygen budget *we chose an online and strictly closed budget approach. Trinh et al. (2024) demonstrated that this approach can yield substantially higher accuracy than offline calculations, especially in the quantification of lateral fluxes.*

*Specific technical comments*

*6) Another aspect that should be clarified concerns oxygen solubility and seasonal variability. Beyond this, I would also recommend showing surface oxygen saturation (%) in the figures, as it would help interpret the observed patterns more effectively. It is expected to observe undersaturation during winter and oversaturation during summer, primarily due to temperature variations: as temperature changes rapidly, oxygen solubility adjusts more slowly, preventing immediate re-equilibration. This process is clearly discussed in Ulses et al. (2021), where the authors attribute it to temperature-driven solubility effects. In contrast, in your manuscript (lines 502–505), the phenomenon is mainly associated with vertical mixing with underlying waters. I am not suggesting that this explanation is incorrect, but as currently written, it seems to imply that mixing is the sole mechanism responsible for these variations.*

**Reply:** The role of temperature-driven oxygen solubility in shaping seasonal surface oxygen variability is discussed in the Results section. In the Discussion, we aimed to underline that wintertime oxygen undersaturation driven by surface cooling is enhanced in water-mass formation areas through the mixing of surface waters with oxygen-poorer subsurface waters. However, we acknowledge that in the discussion the emphasis on vertical mixing may have given the impression that mixing was the sole mechanism involved. To avoid any ambiguity, we have revised this part of the Discussion to explicitly mention the contribution of temperature-dependent solubility changes, in addition to vertical mixing and air–sea gas exchange, and we now refer more clearly to the corresponding Results section. In addition, the surface oxygen saturation (%) will be shown in Figures 4(b) and 8(b), replacing the separate presentation of surface oxygen concentration and oxygen solubility.

Discussion:

*The uptake is enhanced in intermediate water formation areas, in particular in the Rhodes Gyre*  *where undersaturation increases during winter due to surface cooling and mixing of poorer $O_2$ water*

*masses with surface waters, in agreement with what was previously observed and modeled in other water formation areas*

*7) A similar clarification applies to the Subsurface Oxygen Maximum (SOM). Oxygen tends to remain trapped within this layer because a density barrier develops during summer stratification, preventing outgassing. This oxygen is later reintroduced into the surface layers during autumn mixing. This mechanism is, however, inconsistent with your statement that:"During the stratified period, primary production in the surface layer leads to oxygen oversaturation and subsequent outgassing, with a maximum oversaturation of 0.6% observed in summer." The apparent contradiction between the expected seasonal trapping and release of oxygen and the interpretation provided in the manuscript should be addressed and discussed in more detail. Perhaps the vertical layer subdivision used in the analysis should be reconsidered to better capture these seasonal processes (e.g. differences between SOM and surface layer).*

**Reply:** We thank the reviewer for this comment and for highlighting the seasonal trapping and release of oxygen associated with the Subsurface Oxygen Maximum (SOM). We agree that during summer stratification, oxygen can accumulate below the mixed layer due to the development of a density barrier, and can later be redistributed during autumn mixing. However, the coexistence of subsurface oxygen trapping and surface oxygen oversaturation during summer is not contradictory. As shown by Di Biagio et al. (2022), SOM dynamics in oligotrophic systems result from the combined effects of biological production, stratification, and vertical transport, while surface oxygen saturation is primarily modulated by temperature-dependent solubility and air–sea gas exchange. In this framework, summer surface oxygen oversaturation and outgassing can occur simultaneously with subsurface oxygen accumulation under stratified conditions. The statement referring to summer surface oxygen oversaturation in our manuscript specifically concerns the upper surface layer and reflects the combined influence of primary production and temperature-driven solubility changes, consistent with the mechanisms described by Di Biagio et al. (2022) and Ulses et al. (2021). It does not imply that oxygen advected below the mixed layer is directly ventilated to the atmosphere during stratified conditions. We agree that the use of fixed vertical depth layers may partly prevent the distinction between the thin mixed layer and the SOM, particularly during strong stratification. A more refined vertical separation would allow a more detailed SOM-focused analysis, but this lies beyond the scope of the present basin-scale oxygen budget study. However in the text we modified as follows:

*During the stratified period, primary production in the thin mixed layer above the SOM leads to oxygen oversaturation and subsequent outgassing, with a maximum oversaturation of 0.6% observed in summer.*

*8) Another important aspect concerns the use of linear correlations to assess the impact of physical and biogeochemical drivers. It is important to recognize that when two variables share a strong seasonal cycle, their correlation will inevitably appear high, regardless of whether a true causal relationship exists. Moreover, if the variables are not independent, correlation coefficients can be artificially inflated. For instance, in your Figure 2 (which also uses the same float data as in Habibi*

*et al., 2023), you compare modeled and observed surface oxygen and solubility. Since both variables are primarily driven by temperature and thus follow a strong seasonal pattern, a simple regression analysis will necessarily yield high correlations. The same limitation applies to the comparison with Argo data (i.e. Figure 2), which remains difficult to interpret in its current form. As a result, such analyses often provide only a static and potentially misleading view of the relationships between variables. A more robust approach would involve Empirical Orthogonal Function (EOF) analysis to explore the variability associated with different drivers. This method has been successfully applied in Di Biagio et al. (2022), where the authors decomposed the variability of oxygen and related parameters into dominant spatial and temporal modes, and subsequently correlated these modes with environmental drivers. EOFs allow one to isolate the main patterns of variability and to assess their temporal evolution, providing a more dynamic understanding of the underlying mechanisms.*

*Applying EOF analysis would enable you to correlate the principal modes of oxygen variability with specific physical and biogeochemical drivers, leading to more robust and physically meaningful correlations (see Korres et al., 2000; Lionello, P., & Sanna, A. 2005, Pisacane et al., 2006; Alvera-Azcárate et al., 2007; Skliris et al.,2012; Escudier et al., 2021; Menna et al.,2022 ; Di Biagio et al., 2022; and…). Using this type of metric would strengthen your results, as it better captures the underlying variability and allows its evolution to be evaluated over time.*

**Reply:** We thank the reviewer for this detailed and constructive comment and for highlighting the limitations of using linear correlations when variables share a strong seasonal cycle. We agree that correlation coefficients alone do not establish causality and can be inflated when variables are not independent or are driven by a common forcing, such as temperature. However, in the present study, the comparison shown in Figure 2 between modeled and observed surface oxygen concentrations and solubility is not intended to infer causal relationships between variables, but rather to assess the model's ability to reproduce the observed seasonal variability and magnitude of dissolved oxygen along the BGC-Argo float trajectories. The high correlations mainly reflect the coherent seasonal response of both modeled and observed oxygen and solubility, which is a necessary condition for model assessment but is not interpreted here as evidence of mechanistic coupling. We also agree that EOF analysis is a powerful tool to decompose variability into dominant spatial and temporal modes and to relate these modes to physical and biogeochemical drivers, as demonstrated in Di Biagio et al. (2023) and other studies you cited. However, EOF analysis is not well-suited for the specific purpose of the model–observation comparison presented here. In particular, the limited spatial sampling of individual BGC-Argo float trajectories does not allow for a robust EOF decomposition that would be directly comparable between modeled and observed fields. In the current version of the paper, we have calculated the standard deviation and the root mean square difference of both modeled and observed time series to provide a more quantitative assessment of model performance in terms of absolute differences and variability amplitudes between observations and the model. EOF analysis applied to model fields alone could also provide additional insight into the dominant modes of oxygen variability, but such an analysis lies beyond the scope of the present study, which focuses on process-based budget quantification and model–observation consistency. We now explicitly mention

in the Perspectives that future work could benefit from applying variability-based approaches (e.g. EOF or regime-oriented analyses) on model fields to further disentangle the respective roles of physical and biogeochemical drivers across temporal scales.

Conclusion:

*This work represents a first step in our modeling of the dissolved oxygen dynamics in the Levantine Basin. The quasi-permanent Rhodes Gyre dominates the basin-scale oxygen budget, but transient cyclonic and anticyclonic mesoscale structures are expected to contribute to the oxygen variability outside the gyre at shorter time scales, further investigations on the role of the various cyclonic and anticyclonic eddies will be conducted in the future.* *Future work could also benefit from applying variability-based approaches, such as EOF or regime-oriented analyses on model fields, to further disentangle the respective roles of physical and biogeochemical drivers across temporal scales (Di Biago et al., 2023).* *While the 7-year study period provides high-resolution insights into oxygen dynamics, it does not cover long-term climate shifts such as the EMT. Several studies suggest a decadal variability of dissolved oxygen across the whole water column linked to the dense water formations in the south Adriatic and Aegean seas and to the general eastern Mediterranean circulation, notably the reversal of the North Ionian Gyre (Ozer et al., 2020, 2022). A time-extended simulation of the coupled model, in addition to the implementation of a finer vertical resolution at key depths, could contribute to examining this longer variability in the Levantine Basin and the connections between the sub-basins of the eastern Mediterranean. Improving the representation of intermediate-water oxygen dynamics in the Mediterranean is also a necessary step toward better quantifying Mediterranean–Atlantic biogeochemical coupling and its sensitivity to future climate-driven changes in ventilation and circulation.*

*Minor comments / Formal aspects*
*9) From a structural point of view, the manuscript would benefit from substantial streamlining and reorganization. At present, there are too many figures, several of which are not clearly legible, and the amount of text accompanying them appears disproportionately small. Your main Results section spans from line 273 to 455, yet there are 52 lines occupied by captions and subheadings. This leaves approximately 130 lines of text for nine figures in the main text and seven in the supplementary material, an evident imbalance. Overall, the text is somewhat fragmented and hard to follow, partly due to an excessive subdivision into subsections. Additionally, in several cases, the chosen color scales make interpretation difficult.*

**Reply:** In the revised version, we have reorganized the Results section by reducing the number of figures and merging several panels in order to better synthesize the results. In particular, figures presenting closely related diagnostics have been combined, and redundant figures have been removed or moved to the Supplement. In parallel, we have reduced the subdivision of the Results section by merging subsections where appropriate, resulting in a more continuous and coherent narrative. Figure captions have been revised to be more concise, and the accompanying text has been expanded where necessary to better interpret the figures. Finally, color scales have been adjusted to improve legibility and consistency across figures.

*10) The section describing the model setup is overly long. As already highlighted in the manuscript, this model has been validated in numerous previous studies across the Mediterranean; therefore, there*

*is no need to restate these findings. I would suggest condensing everything between lines 185 and 273 into a brief summary of just a few sentences. Furthermore, considering your main scientific goal, to identify and analyze the driver mechanisms, the model comparison should not be treated as a result per se. Instead, it represents a methodological validation step that supports the credibility of your subsequent model-based analyses and process interpretations.*

**Reply:** Following the suggestion of Reviewer 2, we condensed the text from the data description for model assessment to the model evaluation sub-section, and have included the model evaluation sub-section in the material and method section in the revised version. The first sub-section of Material and Method was also shortened.

*11) The Discussion section currently lacks depth and does not adequately address or interpret the defined driver mechanisms; in fact, these mechanisms are not clearly introduced in the Introduction.*

**Reply:** This point is closely related to comments 1 and the remark on extreme events. In response, we have revised the manuscript to better clarify the scope of the study and the mechanisms addressed. In this study we only consider the main processes (air–sea exchange, vertical and lateral transport, and biogeochemical fluxes) the results are explicitly interpreted in terms of these drivers.

*12) Regarding references, this section requires careful revision. For example, when you state: "underlying intermediate layer from 150 to 400 m where LIW flows, and the deep layer below 400 m (Estournel et al., 2021)." This is a modeling reference, which is perfectly valid, but it may not be the most appropriate justification for defining layer boundaries. You could instead complement it with observational or climatological references (for instance, from the SeaDataNet Mediterranean Temperature and Salinity climatology). Alternatively, you could justify your choice explicitly by stating that: "The vertical layers were defined according to the thermohaline structure identified by the physical model." Additionally, several recent and relevant studies for the basin are missing. For example, Pirro et al. (2024) highlights the strong coupling between Ionian surface circulation and mesoscale structures in the Levantine Basin; Velaoras et al. (2017) discusses dense water formation events in the Cretan area. More recent biogeochemical process studies should be considered, as they rely on improved datasets compared to older works. You might also consider adopting the standardized acronyms proposed in Schroeder et al. (2024) for the Mediterranean Sea, which would help maintain consistency with current literature.*

**Reply:** We agree that the definition of the vertical layers should be more clearly justified. We have revised the text to explicitly state that the vertical subdivision is based on the thermohaline structure represented by the physical model and the associated dominant biogeochemical processes. We have also revised the reference list to include more recent and relevant studies for the Levantine and eastern Mediterranean basins, and we have updated acronyms where appropriate to improve consistency with recent literature.

*2.1.3 Budget calculation*

*The water column was divided into three layers based on the*  *thermohaline structure represented by the physical model and the associated dominant biogeochemical processes: the surface layer defined as the photic layer covering the surface to 150 m depth where photosynthesis takes place, the underlying intermediate layer from 150 to 400 m where LIW flows, and the deep layer below 400 m (Estournel et al., 2021).*

*Specific comment*

*Abstract: Line 1 – It's generally better to avoid expressions such as "for the first time." They don't add much value, especially considering that several studies have already addressed oxygen dynamics in this area — we're not exactly at Nielsen's times anymore.*

**Reply:** The expression "for the first time" was removed.

*Lines 27–29 – This statement seems to contradict the classic pattern of the Mediterranean zonal circulation.*

**Reply:** This point is addressed in detail in our response to Comment 3, where we clarify the distinction between lateral oxygen transport and the classical zonal circulation of Levantine Intermediate Water.

*Lines 30–31 – The concept expressed here appears inconsistent with what you mention in lines 25–26. Clearly, both processes occur during the year, but this needs to be rewritten considering seasonality.*

**Reply:** We have revised the text to explicitly distinguish between the seasonal atmospheric oxygen uptake that is dominant in winter and the biological oxygen production in the surface layer, clarifying that these processes occur at different times of the year. In addition, we now describe the seasonal variability of these processes more clearly.

Abstract:

*The model shows that on an annual scale, the basin acts as a net sink for atmospheric oxygen. The surface layer (0-200m) also serves as a source of dissolved oxygen for intermediate depths. Oxygen is transported laterally into the basin from the Ionian Sea and exported towards the Aegean Sea, with winter heat loss intensity enhancing this lateral export at both surface and intermediate layers. The Levantine Basin alternates between autotrophic and heterotrophic states on an annual scale, depending on the intensity of winter surface heat loss. Spatially, the Rhodes Gyre, a quasi-permanent cyclonic structure and major site of intermediate water formation, emerges as a significant oxygen pump in winter, with annual uptake rates twice as high as the rest of the basin, and shows enhanced biological production in the surface layer during the productive season, contributing to 41% of the net annual oxygen production in the surface layer in the Levantine basin.*

*Introduction Overall, the introduction doesn't follow a clear informational structure that moves from a general introduction to the specific context of your study. Instead, it shifts somewhat abruptly between topics. You don't necessarily need to follow a rigid scheme, but it's important to avoid repetition and ensure a smooth, logical flow of ideas.*

**Reply:** We have carefully revised the Introduction to improve its overall structure and logical flow. The specific changes addressing these points are detailed in our responses to the line-by-line comments below.

*Lines 36–40 – There's a lack of key references in this section, particularly regarding how oxygen influences biogeochemical cycles and the main drivers of oxygenation. The latter point should be expanded with recent literature (e.g. review articles) to help readers understand the mechanisms at play; listing the drivers alone isn't sufficient.*

**Reply:** Following reviewer 2's suggestion, we have revised this part of the introduction and have added several recent studies to support this discussion.

Introduction

*Dissolved oxygen (O2) is essential for marine life, supporting respiration of living organisms and the oxidation of organic matter, thereby regulating nutrient cycling and organic matter remineralization, and influencing the biogeochemical cycles of important elements in the ocean (Breitburg et al., 2018; Gruber 2011; Morée et al., 2023). The ocean's oxygen inventory is primarily controlled by photosynthesis, respiration of organic matter, remineralization, temperature and salinity-dependent oxygen solubility, air-sea exchange and the mixing and advective fluxes influencing the ventilation of water masses (Sanders et al. 2026; Helm et al, 2011). Since 1960, the total oxygen inventory has decreased by 2% in the Global Ocean (Schmidtko et al., 2017).*

References:

Breitburg, D., Levin, L. A., Oschlies, A., Grégoire, M., Chavez, F. P., Conley, D. J., ... & Zhang, J. (2018). Declining oxygen in the global ocean and coastal waters. *Science*, *359*(6371), eaam 7240.

Gruber, N.: Warming up, turning sour, losing breath: ocean biogeochemistry under global change, Philos. T. Roy. Soc. A, 369, 1980–1996, https://doi.org/10.1098/rsta.2011.0003, 2011.

Morée, A. L., Clarke, T. M., Cheung, W. W., and Frölicher, T. L.: Impact of deoxygenation and warming on global marine species in the 21st century, Biogeosciences, 20, 2425–2454, https://doi.org/10.5194/bg-20-2425-2023, 2023.

Sanders, R. N. C., McDonagh, E. L., Lauvset, S. K., Turner, C. E., Haine, T. W. N., Goris, N., and Sanders, R.: Remineralisation changes dominate oxygen variability in the North Atlantic, Ocean Sci., 22, 225–240, https://doi.org/10.5194/os-22-225-2026, 2026.

*Lines 41–48 – This part feels too long and lacks recent references.*

**Reply:** We thank the reviewer for this comment. We have shortened this paragraph to improve readability.

~~This decrease in oxygen inventory referred to as deoxygenation has been attributed to the global warming which leads to the reduction of oxygen solubility, explaining ~15% of the current total global oxygen loss (Schmidtko et al., 2017)and the increase of upper ocean stratification generating a reduction of ventilation and circulation of deep ocean layers (Helm et al., 2011; Schmidtko et al., 2017).~~ Global ocean deoxygenation has been

primarily attributed to warming-induced reductions in oxygen solubility and to enhanced upper-ocean stratification, which limits vertical ventilation (Helm et al., 2011; Schmidtko et al., 2017; Stramma and Schmidtko, 2021). However, oxygen changes present large regional and temporal variability (Schmidtko et al., 2017).  particularly at seasonal to decadal time scales, making long-term trends difficult to detect in the upper ocean. Identifying the relative contribution of physical and biogeochemical drivers is therefore essential to better understand regional oxygen dynamics.

*Lines 49–50 – Why are the study objectives presented before the section on the Mediterranean context?*

**Reply:** The study objectives have been moved and are now presented after the section describing the Mediterranean context, in order to improve the logical flow of the Introduction.

*Lines 51–62 – This section is rather brief and also lacks up-to-date references (for instance, I don't see why the work by Taillandier et al., 2022…). More importantly, it omits a crucial aspect when discussing the Eastern Mediterranean, the decadal variability of the NIG, which you mention later in the conclusions and should introduce earlier, and discuss it in the Discussion section.*

**Reply:** We thank the reviewer for highlighting the importance of the NIG variability in the Eastern Mediterranean circulation. We agree that the NIG plays a key role in modulating basin-scale exchanges and water-mass pathways on decadal timescales. We will revise this paragraph by adding recent and lacking references, in particular on the role of NIG on thermohaline properties and LIW formation The decadal variability associated with NIG reversals is not resolved in our simulation over the simulation period (2013–2020) and thus cannot be linked to the oxygen budgets quantified here. For this reason, the reproduction of NIG-related processes will be discussed as a limitation of our simulation and oxygen budget in the Discussion section and are addressed in the conclusions (lines 559-564) where they are framed as an improvement perspective for future work. We believe that this placement appropriately reflects both the importance of the NIG and the specific temporal focus of the present study.

*Line 67 – "The vertical distribution…" – what exactly do you mean here? It seems you're referring to the first ~100 m of the water column, but this should be clarified or rewritten for precision.*

**Reply:** We agree that the wording was imprecise. We have revised the sentence to explicitly indicate that this description refers to the upper water column (surface layer), and we now specify the corresponding depth range to avoid ambiguity.

*The vertical distribution of dissolved oxygen in the upper water column in the basin is characterized by a surface layer (0-100m) exhibiting seasonal variability.*

*Lines 68–71 – You mention that during summer stratification there is a supersaturated surface layer and the SOM, and this two are related to processes such as biological production (which makes sense) and subduction. Are you sure the surface supersaturation is not instead driven mainly by temperature-related changes in solubility?*

**Reply:** We agree that surface oxygen supersaturation during summer stratification is largely driven by temperature-dependent solubility changes. We have revised the text to explicitly mention the temperature-dependent solubility changes leading to surface supersaturation.

*Lines 72–73 – Could you clarify what you mean by "related to the atmosphere"?*

**Reply:** We have clarified this sentence to explicitly state that surface oxygen undersaturation refers to undersaturation relative to atmospheric equilibrium, driven by low winter surface temperatures, rather than using the expression "related to the atmosphere".

*In winter, the oxygen vertical profile shows an upper mixed layer with maximum dissolved concentrations of 240 μmol kg$^{-1}$, characterised by an undersaturation in oxygen at the surface relative to atmospheric* equilibrium, promoting atmospheric oxygen uptake.

*Lines 74–75 – The oxygen minimum in the Levantine Basin oscillates in phase with the NIG rotation.*

**Reply:** To our knowledge, there is no direct observational or modeling study demonstrating that the oxygen minimum layer in the Levantine Basin oscillates *in phase* with the North Ionian Gyre rotation. What is well established is that the cyclonic–anticyclonic reversals of the NIG (BiOS mechanism) strongly modulate the advection of different water masses of Levantine and Atlantic origin (e.g. Gačić et al., 2010; Manca et al., 2006; Menna et al., 2022), thereby influencing salinity and biogeochemical properties, including nutrient availability, oxygen vertical distribution and ecosystem dynamics (Civitarese et al., 2010; Batistić et al., 2014; Mauri et al., 2021; Ozer et al. 2022). While such circulation variability may indirectly affect dissolved oxygen through changes in water mass properties and ventilation pathways, a direct phase-locked oscillation of the oxygen minimum with the NIG rotation has not been clearly documented.

*Line 78 – As written, this statement sounds speculative. However, if you explain it and link it to NIG variability, the point would become much stronger and more meaningful.*

**Reply:** The statement at Line 78 is intended as a general introductory description of the observed temporal and spatial variability of dissolved oxygen in the Levantine Basin, as documented by Kress et al. (2014), and serves to introduce the paragraph. It is not meant to imply a specific or exclusive driving mechanism. We agree that decadal-scale oxygen variability in the Eastern Mediterranean can be influenced by large-scale circulation changes, including variability associated with the North Ionian Gyre and the BiOS system (e.g. Gačić et al., 2011; Borzelli and Carniel, 2023). In particular, the study by Ozer et al. (2022) suggests a possible influence of the BiOS on oxygen content in the LIW through changes in the thermal and biogeochemical properties of water masses flowing into the Levantine Basin. However, establishing a direct or phase-locked relationship between NIG variability

and oxygen changes in the Levantine Basin would require a dedicated analysis beyond the scope of the present study. To clarify this point and avoid any ambiguity, we have slightly revised the sentence to acknowledge the potential role of large-scale circulation variability, without implying a direct causal linkage. We have revised the sentence to clarify its introductory nature and to acknowledge the potential role of large-scale circulation variability, and to explicitly refer to the study by Ozer et al. (2022).

Introduction

*The Levantine Basin shows spatial changes in oxygen content occurring at annual and decadal time scales (Kress et al., 2014; Sisma-Ventura et al., 2016), reflecting the combined influence of biogeochemical processes and large-scale circulation variability.*

References:

Eusebi Borzelli, G. L., & Carniel, S. (2023). *A reconciling vision of the Adriatic–Ionian Bimodal Oscillating System*. Scientific Reports, 13, 2334. https://doi.org/10.1038/s41598-023-29162-2

Gačić, M., Civitarese, G., Eusebi Borzelli, G. L., Kovačević, V., Poulain, P.-M., Theocharis, A., Menna, M., Catucci, A., & Zarokanellos, N. (2011). *On the relationship between the decadal oscillations of the northern Ionian Sea and the salinity distributions in the eastern Mediterranean*. Journal of Geophysical Research: Oceans, 116, C12002. https://doi.org/10.1029/2011JC007280

*Material and Methods: In my opinion, it's not necessary to re-explain concepts that have already been well described in previous studies.*

**Reply:** We thank the reviewer for this comment. We have taken this remark into account and have compacted the manuscript by shortening sections where concepts previously described in earlier studies were reiterated, while keeping only the elements necessary.

*Section 2.1.3 – I don't see why the study area needs to be reintroduced here if it has already been described in the introduction, and also why it's presented together with the budget section.*

**Reply:** We thank the reviewer for this remark. In this section, the study area is briefly redefined in order to explicitly specify the spatial domain used for the oxygen budget calculations. The section title has been revised and is now "2.1.3 Budget calculation", clarifying that this subsection focuses on the methodological definition of the budget domain rather than on a general presentation of the study area.